



# Forest structure and individual tree inventories of north-eastern Siberia along climatic gradients

Timon Miesner[1,2], Ulrike Herzschuh[1,2,3], Luidmila A. Pestryakova[4], Mareike Wieczorek[1], Evgenii S. Zakharov[5,4], Alexei I. Kolmogorov[4], Paraskovya V. Davydova[4], and Stefan Kruse[1]

[1]Alfred Wegener Institute Helmholtz Centre for Polar and Marine Research, Telegrafenberg A45, 14473 Potsdam, Germany
[2]Institute of Environmental Sciences and Geography, University of Potsdam, 14476 Potsdam-Golm, Germany
[3]Institute of Biochemistry and Biology, University of Potsdam, 14476 Potsdam-Golm, Germany
[4]Institute of Natural Sciences, North-Eastern Federal University, 677000 Yakutsk, Russia
[5]Institute for Biological Problems of Cryolithozone, 677000 Yakutsk, Russia

**Correspondence:** Timon Miesner (timon.miesner@awi.de)

**Abstract.** We compile a data set of forest surveys from expeditions to the north-east of the Russian Federation, in Krasnoyarsk Krai, the Republic of Sakha (Yakutia) and the Chukotka Autonomous Okrug (59-73° N, 97-169° E). The region is characterized by permafrost soils, and forests dominated by larch (*Larix gmelinii* Rupr., *Larix cajanderi* Mayr).

Our dataset consists of a plot data base describing 226 georeferenced vegetation survey sites, and of a tree data base with
information about all trees on these plots. The tree data base contains information on height, species and vitality of 40,289 trees. A subset of the trees was subject to a more detailed inventory, recording stem diameter at base and at breast height, crown diameter and height of the beginning of the crown.

We recorded heights up to 28.5 m (median = 2.5 m) and stand densities up to 120,000 trees per ha (median = 1197 ha$^{-1}$), both values tending to be higher in the more southerly areas. Observed taxa include *Larix* Mill., *Pinus* L., *Picea* A.Dietr.,
*Abies* Mill., *Salix* L., *Betula* L., *Populus* L., *Alnus* Mill. and *Ulmus* L..

In this study, we present the forest inventory data aggregated per site. Additionally, we connect it with different remote sensing data products to find out how accurately forest structure can be predicted from such products. Allometries were calculated to obtain the diameter from height measurements for every species group. For *Larix*, the most frequent of ten species groups, allometries depend also on the stand density, as denser stands are characterized by thinner trees, relative to height. The remote
sensing products used to compare against the inventory data include climate, forest biomass, canopy height, and forest loss or disturbance. We find that the forest metrics measured in the field can only be reconstructed from the remote sensing data to a limited extent, as they depend on local properties. This illustrates the need for ground inventories like those data we present here.

The data can be used for studying the forest structure of north-eastern Siberia, and for the calibration and validation of
remotely sensed data.





# 1 Introduction

Twenty percent of the world's forests are located in Russia (FAO 2020), much of these in the sparsely populated north and east of the country. As the high latitudes are warming at a much faster rate than the global average, these forests are experiencing and will face further massive, abrupt changes (Scheffer et al. 2012). The threat of feedback loops to the global climate system

(Bonan 2008), possibly through the thawing of permafrost (Schuur et al. 2015) or changes in biosphere and soil carbon stocks (Walker et al. 2019), make it crucial to understand these ecosystems.

While the major portion of the world's boreal forests are made up of evergreen coniferous forest, north-east Asia is dominated by summergreen coniferous trees of the species *Larix gmelinii* and *Larix cajanderi* (Abaimov 2010). This vegetation type covers an area of several million square kilometers and stretches from northern China in the south and the Central Siberian Plateau

in the west, where mixed stands occur with evergreen coniferous trees, to the northern treeline near the Arctic Ocean, where sparse forest tundra and stunted growth forms prevail (Wieczorek et al. 2017; Kruse et al. 2020). Much of the geographical range is underlain by continuous permafrost (Osawa et al. 2010). Recurrent forest fires also play a vital role in the ecosystem (Payette 1992).

There has been no comprehensive forest inventory and planning in Russia in the post-Soviet era, and thus estimations on the

volume of wood in the nation's forests vary widely (Schepaschenko et al. 2021). A national forest inventory, conducted between 2006 and 2020, aimed to shed light on this, but no definite results have been published as of May 2022. There are only a few studies that deal explicitly with larch dominated ecosystems in Russia, for example (Kharuk et al. 2019; Dolman et al. 2004) and the comprehensive volume by (Osawa et al. 2010). The range of *Larix gmelinii* extends into the northernmost provinces of China, where it is used for afforestation. In this area, there has been much research on this species, e.g. (Jia & Zhou 2018;

Widagdo et al. 2020; Xiao et al. 2020), but the properties of the species -and thus the ecosystems it forms - vary widely depending on growing conditions, which are a lot harsher in the northern parts of its range (Wang et al. 2005).

Remote sensing data can give insights into many forest-related parameters, such as above-ground biomass, growing stock volume or canopy height (Simard et al. 2011, Santoro et al. 2018), and in the past decade, there has been a massive increase in detailed, freely available remote sensing data products. The ground-truthing that is necessary for such products tends to

have a bias towards more accessible forest areas, where previous forest surveys have been conducted (e.g. Yang & Kondoh 2020). Another issue is that sparsely forested ecosystems at the tundra-taiga ecotone are often not understood as forests, e.g. by the influential FAO definition (FAO 2000), and therefore they may be excluded from such data. Other aspects, such as the compositional complexity of forest in terms of height, age, and species distribution, can hardly be captured remotely at all, meaning that it is still necessary to take on-site measurements in order to understand these ecosystems.

To meet this demand, joint Russian-German expeditions to Siberia have been conducted since 2011 to the Russian Federation Subjects of Krasnoyarski Krai, the Republic of Sakha (Yakutia), and the Chukotka Autonomous Okrug. In this study, we present the collected forest measurement data of the combined expeditions, both at the level of single trees, and at the plot level, which can potentially be further upscaled. The central questions that motivate this study are: What are the patterns of forest composition in north-east Asian larch ecosystems? How much growing stock of wood do they hold? How strong is the



role of climate as a driver for these variables? How well do available remote sensing products describe what we see on the ground?

## 2    Methodology

### 2.1    Area of interest

The areas of interest are the larch dominated forests in north-east Asia, including the transition zones to the tundra and to

evergreen deciduous forest (see Figure 1). The area is characterized by permafrost soils and extremely continental climate (Kajimoto et al. 1999). Precipitation is generally below 300 $mm$ per year, with exceptions towards the boundaries of the area. Winter temperatures are mostly below -30°C, while the warmest months average between 20°C in central Yakutia to 8°C near the Arctic Ocean. The forests of the region are sparse and slow-growing. Recurring fires are an important driver for this ecological system (Kharuk et al. 2011).

### 2.2    Forest inventories


Eight summer expeditions were led to different destinations in the Russian Federation: to the tundra-taiga transition zone in 2011, 2012, 2013, 2014, 2016, and 2018, to the mountainous tundra treeline in 2016 and 2018, and to the boreal forest in 2018 and 2021) (Overduin et al. 2017; Kruse et al 2019). The main goals varied between the expeditions, but all included forest inventories using the same methodology. The expedition sites are not evenly distributed across the area, as the focus was on

transition zones, especially the tundra-taiga ecotone at the northern limit of the range of *Larix*, and the transition to evergreen forests in the south-west of its range.

The sites at which the surveys were performed were chosen beforehand with consideration of remote sensing data. The goal was to cover a wide range of conditions such as tree cover percentage and reflectance values in the region of each expedition. The exact positioning of the plots was finalized on site, with the aim to have each plot representing a homogeneous vegetation

type. Not all vegetation survey plots contain forest or even single trees; some were used to record ground vegetation and tree recruitment, while taller trees were absent.

Plots were either rectangular or circular. Rectangular plots were more commonly used in the tundra-taiga ecotone. They would typically be squares of 20 m x 20 m, but their size was sometimes increased in areas with very few trees per hectare, or decreased in size if vegetation or topography demanded it. A grid of 2 m x 2 m was laid out over the plot in order to

locate trees precisely inside of it. In a rectangular plot, every tree was recorded in detail, noting the following variables: species, height, vitality estimate (on a discrete scale, from "very vital", "vital", "mediocre", "low", "very low" to "dead"), basal diameter, diameter at breast height (DBH), maximum crown diameter, and the smaller crown diameter, which was measured perpendicular to the maximum.

Circular plots had a diameter of 15 m, except for occasions in which the forest was too dense to record all trees in this range;

in these cases, the diameter was reduced to 10 m. They were divided into four quadrants along the cardinal directions. Of the

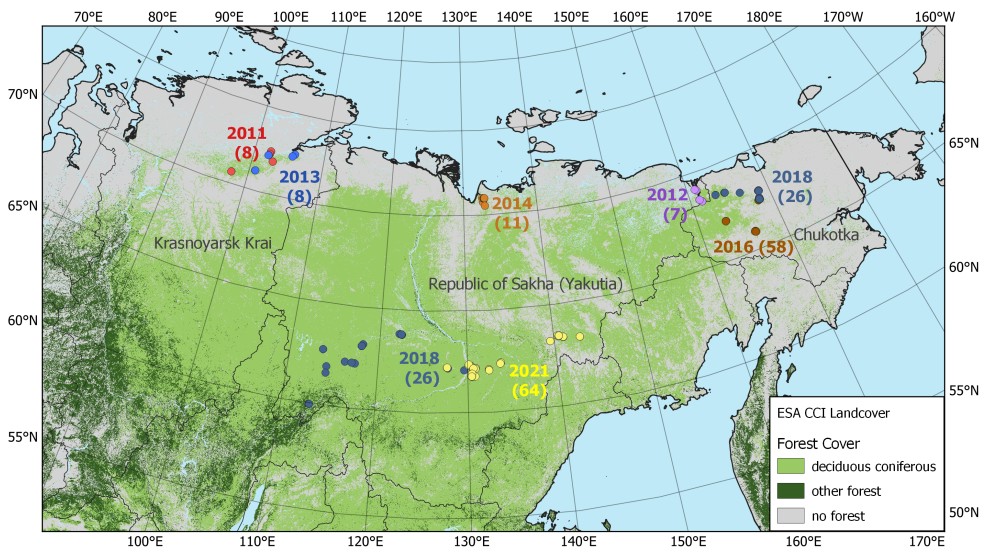

**Figure 1.** The vegetation in the larch-dominated forests of north-eastern Russia. Numbers indicate the year and the number of vegetation plots on each expedition.

trees in the circular plot area, a minimum of 10 trees were chosen for detailed inventory as above. The goal was to choose 10 trees per species so that they covered the entire range of height and diameter variation present on the plot. If there were more than two species, the number of chosen trees per species was reduced due to time constraints, with the focus on coniferous trees. After making the detailed inventory of the chosen trees, all trees on the plot were recorded noting only species, estimated 90 height and general remarks, for example whether the tree had low vitality, was dead, inclined, or not of upright growth form.

Tree height was measured with a clinometer for some trees, and for others visually estimated by making a comparison with the measured trees or objects of known height. According to experience, the error of this method was below 10% for smaller trees or below 1 m for larger ones. Generally, all trees at least 40 cm in height were measured. Additionally, for many sites along the treeline, where recruitment was the focus of the research, smaller individuals were recorded on sub-plots. Stem 95 diameters were measured either with a measuring tape (as circumference) or a calliper, recording the basal diameter just above the root collar and DBH at 1.3 m above the ground. Crown diameters were estimated from below, with the help of ground measurements using a measuring tape.

Parts of the data set presented here, have already been published in other data publications and are available individually:





- Wieczorek et al. 2017: *Field and simulation data for larches growing in the Taimyr treeline ecotone* (including data of 2011 and 2013 expeditions); DOI: 10.1594/PANGAEA.874615

- Kruse et al. 2020: *Forest inventories on circular plots on the expedition Chukotka 2018, NE Russia*; DOI: 10.1594/PANGAEA.923638

- van Geffen et al. 2021: Tree height and crown diameter during fieldwork expeditions that took place in 2018 in Central Yakutia and Chukotka, Siberia; DOI: 10.1594/PANGAEA.932817

## 2.3 Processing of the data

In the tree database, every entry contains information about one tree. Some processing was done prior to analysis, to derive variables that were not present in the original dataset.

The species of each tree was recorded differently depending on the surveyor. This led to differences in the naming convention, for example *Betula pendula* on some sites, and *Betula spec.* on others. Therefore, the 23 taxa entries were harmonized into 110 ten species groups, identified by the genus name. The species *Larix gmelinii* and *Larix cajanderi* were grouped together in the species group *Larix*. An exception is the genus *Pinus*, where *Pinus pumila* ((PALL.) REGEL) was excluded from the *Pinus* group due to its shrub-like growth form.

As height was recorded for all trees, but diameters only for selected ones, the existing diameters were used to calculate allometries, from which the diameters were then reconstructed from the height for those trees where they were not measured. For each species group, a power function of the form

$$D_{BS} = a_1 \cdot H^{a_2}$$

was fitted with the least squares method, (where $D_{BS}$ is the diameter at base, $H$ is the height, and $a$ and $b$ are the optimization coefficients). For diameter at breast height ($D_{BH}$), the function is:

$$D_{BH} = a_1 \cdot (H - 1.3)^{a_2}$$

Initial analyses with this function revealed that the diameter estimations were biased on some plots: On densely forested sites, trees tended to have smaller diameters at the same height compared to sparsely forested plots, especially in the lower half of the height range. As the power functions computed for the different stand density groups (measured in trees per ha) differed both in exponent and in factor, we used the adjusted power function

$$D_{BS} = (a_1 + a_3 \cdot S) \cdot H^{(a_2 + a_4 \cdot S)}$$

where the stand density $S$ was computed from $T_{ha}$, the number of trees per hectare, as follows:

$$S = \max(\log_{10}(T_{ha}), 2)$$





The formula for $D_{BH}$ was analogous, replacing $H$ by $(H - 1.3)$. The latter formulas were only applied to the species group *Larix*, as all other species were not present on enough different plots to prevent overfitting. For all other species groups, the

former, simpler formulas were used.

Having thus obtained the variables predicted DBS and predicted DBH for all trees, it was possible to calculate further metrics, including basal area $(BA)$ as

$$BA = \frac{\pi}{4} D_{BH}{}^2$$

and stem volume $(V)$, which was obtained using the Smalian volume formula (Cailliez & Alder 1980) for trees taller than breast height

$$V = \frac{D_{BS}{}^2 + D_{BH}{}^2}{2} \cdot \frac{\pi}{4} \cdot 1.3 + \frac{D_{BH}{}^2}{2} \cdot \frac{\pi}{4} \cdot (H - 1.3)$$

and respectively for trees smaller than breast height, and

$$V = \frac{D_{BS}{}^2}{2} \cdot \frac{\pi}{4} \cdot H$$

After calculating these variables for the individual trees, they were aggregated at the plot level by calculating mean and selected quantiles of height as well as sum of basal area and stem volume. The latter variables were then divided by the plot area, to get the respective values per hectare.

Another measure we calculated for the height distributions of each plot is the Gini coefficient (Gini 1912). It ranges between 0 and 1, assuming a value 0 if all trees have the same height, and approaching 1 if there are a few very big trees alongside many very small ones. Let $h_i$ be a collection of height measurements in ascending order, and $i$ in $\{1, ..., n\}$, then the Gini-Coefficient is defined as

$$1 - 2\frac{\sum_{i=1}^{n}(h_i \cdot (n - i + 0.5))}{\sum_{i=1}^{n}(h_i \cdot n)}$$

**2.4   Remote sensing data for comparison**

In the study, we used remote sensing derived data products on climate, biomass, height, forest cover loss, and stand age to compare with and relate to the forest inventory.

**2.4.1   Climate**

CHELSA - "Climatologies at high resolution for the Earth's land surface areas" (Karger et al. 2017; Karger et al. 2021) is a

global raster dataset containing many different variables. This study uses the monthly temperature means and monthly precipitation values, as well as the bioclimatic variables mean anual temperature, diurnal temperature range, temperature seasonality, growing degree days above 0°C (GDD0), above 5°C (GDD5) and above 10°C (GDD10), length of the growing season (GSL), mean maximum temperature of the warmest month and mean minimum temperature of the coldest month, and the first and last day with temperatures above 0°C, above 5°C and above 10°C. This made a total of 46 climate variables.

All values are means for the period 1981-2010, with a spatial resolution of 30 degree seconds - less than 1 km.

### 2.4.2 Forest biomass

The GlobBiomass dataset (Santoro et al. 2018a) covers theEearth's land surface with a pixel size of one hectare. It provides values for above ground biomass (AGB) and growing stock volume (GSV) for the year 2010, as well as the standard errors, derived from satellite-based synthetic aperture radar, and an extensive set of ground measurements. The authors note that their data set is not precise at the pixel-level, but only over larger areas.

### 2.4.3 Forest height

The forest canopy height product (Simard et al. 2011) is a raster data set with a resolution of 1 km$^2$. It estimates the maximum canopy height in each pixel from the GLAS satellite-borne lidar, using additional data about climate, elevation, and canopy cover. All values are for the year 2005.

### 2.4.4 Tree cover loss

We used the tree cover loss product from the Global Forest Watch project (Hansen et al. 2013) which is based on yearly observations of Landsat images; therefore the spatial resolution equals that of Landsat with 30m. The project has published various related data sets which are updated regularly, such as forest cover for any given year between 2000 and 2019, and tree cover gain per year. The tree cover loss product is thus derived from the annual forest cover products, assigning the year of the loss to a given pixel, or 0 if no loss has taken place since the year 2000.

### 2.4.5 Siberian larch stand age

Distribution of Estimated Stand Age Across Siberian Larch Forests (Chen et al. 2017) is related to the former dataset, and is also mainly based on Landsat images with 30m resolution. It incorporates some more analysis to detect stand-replacing forest fires, but it only covers a part of eastern Siberia, including, however, 54 of our vegetation survey plots, and spans the years 1989-2012. For every pixel, it gives the age of the forest stand if it has experienced a stand replacing fire since 1989, a value 100 if there has been no fire 1989-2012, or no data if the pixel does not contain larch forest.

### 2.5 Analysis methods

The remote sensing products that were used all consisted of raster data. The values at the locations of the plot centres were extracted using QGIS 3.16.

The CHELSA climate data set with its 46 climatic variables for the 226 survey plots was subjected to a principal component analysis (PCA). Subsequently, a subset of the variables was chosen for further analyses, namely "annual precipitation sum" (Prec.), "January mean temperature" (T01), "July mean temperature" (T07), and "growing degree days above 0°C" (GDD0). Univariate linear regressions were calculated between every single variable and four forest inventory variables, as well as multilinear regressions beween all the climate variables and the same forest variables.



160    To compare the GlobBiomass product and the Forest Height product with our data, linear regressions were calculated between the remote sensing -derived variables and suitable variables of our forest plot data.

We compared the quotient of living basal area over total basal area for sites with recent tree cover loss and sites without recent tree cover loss as assessed by a two-sided t-test.

All analysis was performed in R 4.1.0 (R Core Team 2021).

## 3   Results

### 3.1   Description of the data

#### 3.1.1   Descriptive statistics

The tree database comprises 42675 entries, describing 40289 trees. This is due to the fact that on circular plots, the trees that were subject to detailed inventory are also recorded again in the height-only inventory. Of these, 33513 individuals were used for aggregation at the plot level. The rest were excluded for being smaller than 40 cm because such trees were not recorded on every plot, or for being located outside of the vegetation plots listed in the plot database.

The plot database includes 226 vegetation plots, of which only 162 contain trees taller or equal to 40cm, while 60 do not. Of the 40289 trees, 4660 were dead, and 35629 living at the time of recording. All entries in the tree database have a recorded height, which ranges up to to 28.5 m. The species is recorded for all but 31 entries. The most frequent species are *Larix cajanderi* (44.4% of database) and *Larix gmelinii* (25.7%). They never occur together on the same plot. Other frequent taxa are *Betula pendula* ROTH (13.9%), *Picea obovata* LEDEB. (5.8%), *Pinus sylvestris* L. (5.0%) and the genus *Salix spec.* (3.2%). Among the less frequent are *Populus tremula* L., *Alnus spec.*, *Pinus pumila* REGEL, *Pinus sibirica* DU TOUR, and *Abies sibirica* LEDEB..

Values for basal diameter are only present for 2583 entries. They range from 0 up to 97.7 cm, with median 6.99 cm and mean 11.08 cm. For diameter at breast height (DBH), there are 2095 values in the dataset, almost all of which are trees for which basal diameter is also given. DBH is almost always lower than basal diameter, on average by the factor 0.628. It ranges up to 71.6 cm, with median 6.4 cm mean 9.02 cm. Maximum crown diameter and smaller crown diameter (measured perpendicular to maximum) are given for 2079 entries, and range from 0 to 16 m. The quotient of the two diameters is, on average, 0.81. Tree crown area, which is the product of the two values and the factor $\frac{\pi}{4} \cdot \frac{1m^2}{10000cm^2}$, is, on average, 4.77 m$^2$, with a median 1.43 m$^2$.

#### 3.1.2   Diameter-height allometry

The power function allometries for the different species differ notably, as can be seen in figure 2. The basal diameter of birches (*Betula*), for example, is obtained from height with an exponent of $a_1 = 1.15$ and factor of $a_2 = 0.91$, while for *Abies*, the exponent is $a_1 = 0.66$, and the factor $a_2 = 2.69$. The genus *Populus* differs strongly from the other species groups, with an exponent of $a_1 = 2.29$ and a factor of $a_2 = 0.06$. In the DBH-model *Populus* differs remarkably from the others, too, even if not that strongly. All factors and exponents are displayed in Appendix B.



The last graph of figure 2 shows the diameter-height allometries for the genus *Larix* when taking into account the number of trees per hectare. When tree measurements are grouped by stand density, the resulting power functions differ by more than the respective standard errors for the coeficcients, especially for heights between 4 and 12 m, where a higher number of trees on the plot have smaller diameters.

### 3.1.3 Height distributions

Tree heights show a nearly exponential distribution, with the exception that values from approximately 15m upward occur slightly more frequently than expected under an exponential distribution (figure 3). However, at the level of individual plots, the distribution patterns vary widely. This can be seen in figure 4: although tree heights on plot EN21-260 are close to an exponential distribution, suggesting a continuous recruitment rate, in EN21-253 the larger trees are over-represented. Plot EN21-230 is missing the smallest cohort, and plot EN21-246 is an example of dense regrowth after a stand-replacing fire, where older trees taller than 7m are absent. Plot EN21-226 is dominated by a cohort of middle-sized trees, lacking both small and very large ones. In EN21-219, some large and many small individuals are present, while medium-sized ones are missing.

The Gini coefficient is normally distributed with a mean of 0.363 and standard deviation of 0.123. Plot EN21-258 is an example of a plot with a high Gini value (0.679), and plot EN21-226 is at the lower end with a Gini coefficient of 0.166.

### 3.1.4 Species distribution

In accordance with the known ranges of the different species, we observe that species diversity tends to be higher on the plots in central and western Yakutia, which experience warmer summers and longer growing seasons than the plots near the northern tree line. All plots north of 70° N have only one species (larch), while on the plots south of 65° N, there are, on average, 3.11 species, with a maximum of 9 tree species from 7 species groups.

The species *Pinus sylvestris, Picea obovata, Abies sibirica, Ulmus spec.* and *Populus tremula* only occur on the sites south of 65° N with a July temperature of at least 17°C. More predominant among the southerly sites are *Betula pendula* and *Alnus spec.*, but they are also found at one, and three sites, respectively, in Chukotka. More frequent between 65° N and 70° N are *Pinus pumila* and *Salix spec.*. Of the plots with trees, all but one have *Larix* individuals. On the sites west of 130°E, it is *L. gmelinii*, and on the sites east thereof, *L. cajanderi*.

### 3.2 Remote sensing products as predictors

#### 3.2.1 CHELSA Climate

The principal component analysis on the data reveals that 95.3% of the variance is captured in the first component, and 99.6% in the first three. The climate on the plots is strongly continental (figure 6), with mild to warm summers, and extremely cold winters. The length of the growing season is between 63 and 132 days, and GDD0 ranges from 565 to 1974.

Weak correlations between four climate parameters (precipitation, January temperature, July temperature, GDD0) and four forest structure parameters (mean height, $\log_{10}$(number of trees per ha), basal area per ha, and stem volume per ha) are found



**Figure 2.** Diameter at base (left) and Diameter at breast height (right) against height, per species. Power function allometries per species shown. Bottom: *Larix* only, coloured by trees per ha, with allometries for four different stand density groups.

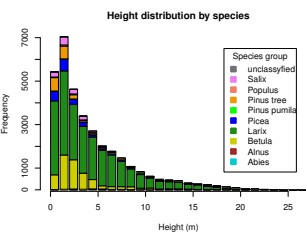

**Figure 3.** Height distribution of all trees on plots

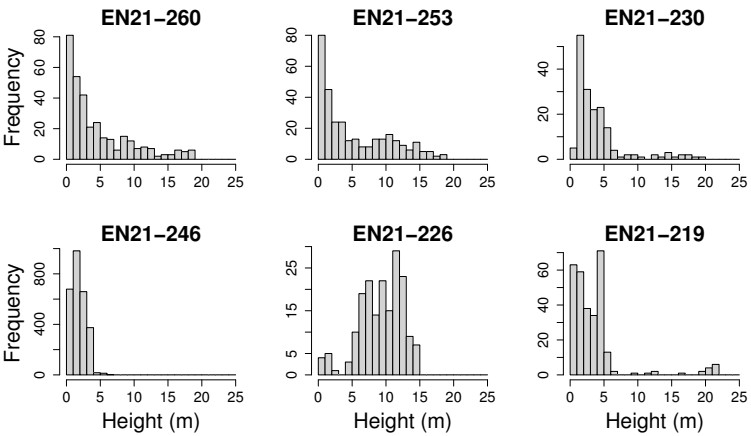

**Figure 4.** The height classes among all species, for six different plots of the Yakutia 2021 expedition.

(Figure 7). The climate variables mean January temperature (T01) and precipitation have very low correlation coefficients with all forest metrics. The correlations between T01 and the forest metrics are even negative, although $R^2$ values are close to 0. Mean July temperature (T07) and GDD0, which themselves are highly correlated ($R^2 = 0.983$), are more strongly correlated

225 with several forest structure parameters, but the strength of the correlation is only intermediate, not exceeding 0.351 in any combination.

Multilinear models with all four climate variables do not perform much better: the maximum adjusted $R^2$ becomes 0.356, and over all four target variables, it is at most 0.027 higher than for the most powerful single predictor (T07).

### 3.2.2 GlobBiomass

230 The two leading variables from the GlobBiomass dataset - above-ground biomass (AGB) and growing stock volume (GSV) - are themselves strongly correlated ($R^2 = 0.989$ over all plots), therefore we focus on just one of them - GSV - which can be derived from our data with more confidence, since we did not measure wood density and biomass expansion factors.

Remote sensing-derived GSV and inventory-derived GSV follow the same tendency (correlation with $R^2 = 0.49$ and residual standard error 79.9; Figure 8). But for some plots, the two values differ by more than an order of magnitude.



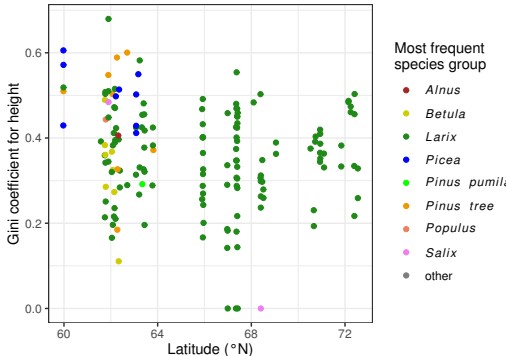

**Figure 5.** Gini coefficients for height, against Latitude, coloured by most frequent species on plot. (Plots with more diverse height distributions have higher Gini coefficients.)

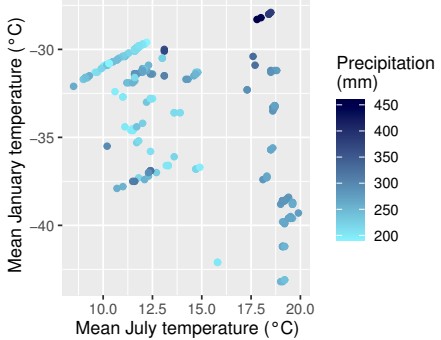

**Figure 6.** The climate on the plots: mean Janurary temperature on the y-axis, mean July temperature on the x-axis and mean annual precipitation as dot colour.

### 3.2.3 Forest height

The values of the Simard et al. (2018) data are 0 (no forest) or integers between 11 and 27 for the forest height in metres. On 125 of the plots, they record a value of 0, while we actually encountered trees on 60 of these plots in our inventory. A linear correlation between Simard canopy height and maximum tree height on the plot (figure 9) has an intercept of 8.55, a slope of 0.298, and $R^2 = 0.20$. Other metrics, such as the 98[th], 90[th] or 75[th] percentiles of the observed tree height, have even less correlation (see ppendix C]).

### 3.2.4 Forest loss

The dataset "Stand Age of Siberian Larch Forests" by ORNL DAAC has data for 54 of our vegetation plots and finds 6 plots have experienced stand-replacing events between 1989 and 2012. The Hansen et al. (2013) dataset disagrees with the former on 5 plots, detecting forest loss in times and places where Stand Age is at maximum.



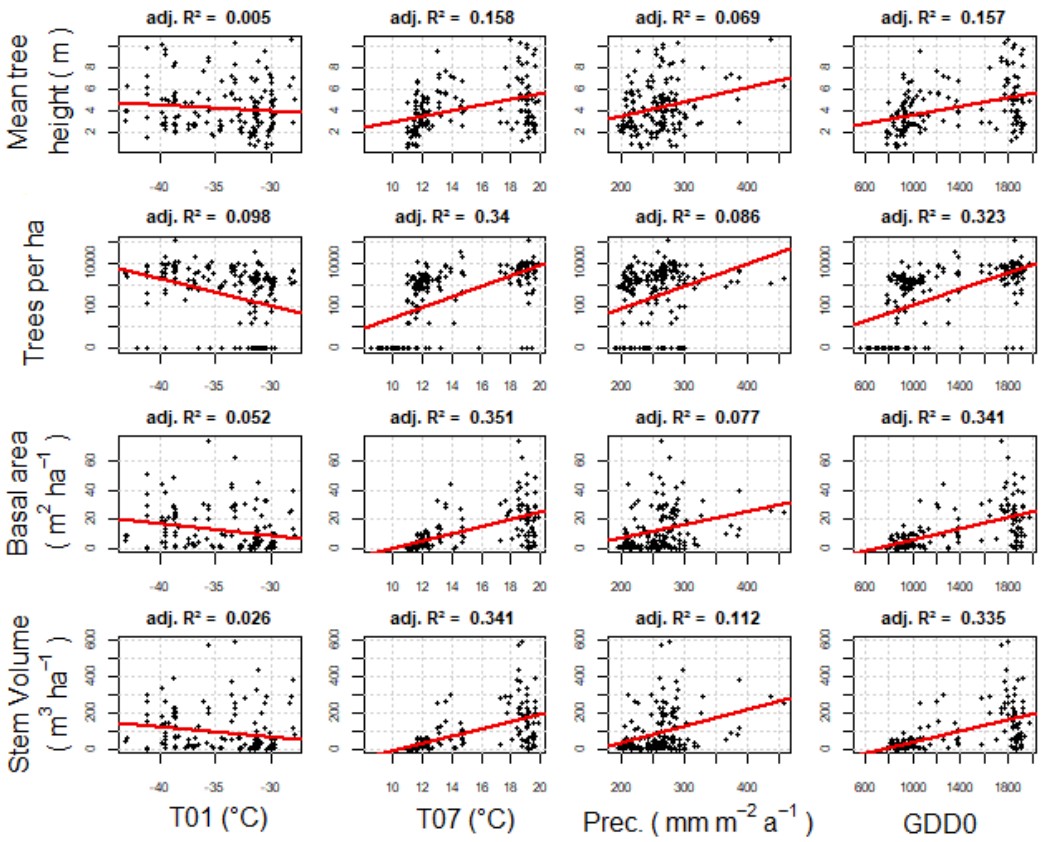

**Figure 7.** Comparison of forest inventory variables with climate variables. Linear regression lines in red.

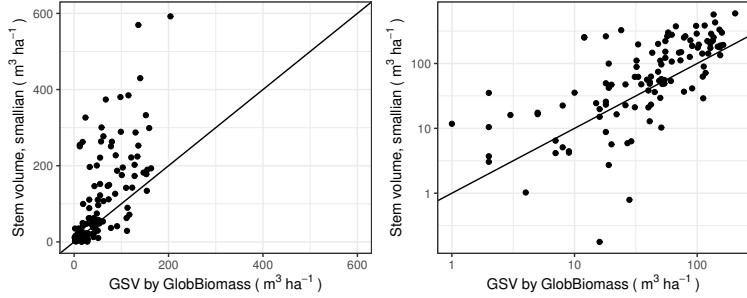

**Figure 8.** Stem volume calculations plotted against growing stock volume (GSV) from the GlobBiomass data set. Left: Linear scale; Right: Logarithmic scale, zeros removed.



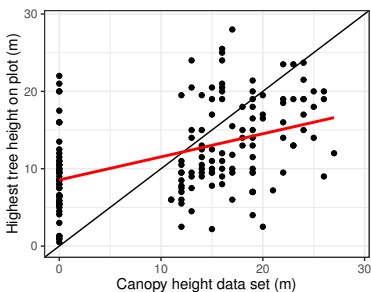

**Figure 9.** Highest tree of the inventory sites plotted against canopy height according to the Simard et al. (2011) data set. Linear regression line in red.

Using only the spatially complete Hansen et al. (2013) data set, we observe that sites with recent forest loss events hold more standing dead trees, measured as the ratio of basal area of living trees to overall basal area (Figure 10). Although a t-test finds that the two groups differ very significantly ($p = 4e-6$), we see that there are also individual plots in which dead trees do not constitute a relevant amount of the basal area.

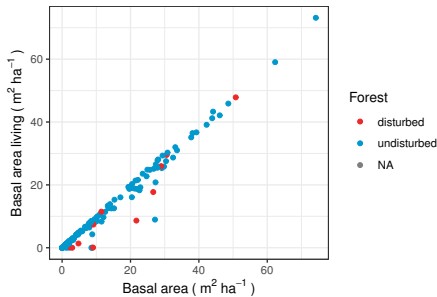

**Figure 10.** Living wood volume compared to overall wood volume; plots with recent forest loss marked red

## 4 Discussion

### 4.1 Relevance of the data set

The data we present in this study are unique in their extent for the regions they cover: (Schepaschenko et al. 2017) have compiled a vast number of forest inventories in Eurasia, but their coverage of our study region is very sparse. For example, they include no data from Chukotka and the Kolyma area, where our data set has 91 sites. The same is true for the validation data set used by (Yang & Kondoh 2020), who have only one location within our area of interest, from more than 400 literature sources they reviewed. This shows the lack of forest inventories from north-eastern Siberia, which our data set aims to mend.



## 4.2 Validity of methods

The field work was carried out according to scientific standards. Tree height was chosen as the leading variable because it is easy to overview in sparse stands and it generally correlates well with other variables (stem diameter, biomass). Frequent clinometer measurements assured precise height estimations, and the remaining errors can be expected to average out over the
high number of observations, which were easily obtained due to the efficiency of the method.

The correlations of the forest metrics with climate variables cannot be generalised, because the distribution of the plots is not representative for the area. Even though the survey plots in each region cover the entire range of vegetation in any given zone, they are not weighed accordingly in the plot data set. However, the relationships can still give us some idea of the general behaviour of the variables.

## 265 4.3 Tree species and heights distribution

We observe a higher species diversity in the more southerly stands, which experience longer, warmer growing seasons. This is in accordance with expectations and the known ranges of the observed tree species (Kuznetsova et al. 2010).

It is uncommon in the literature to record height distributions, but methodological analogues are age-class or diameter distributions, which can be used to show recruitment patterns, e.g. (Lin et al. 2005). While the close-to-exponential distribution
of tree height suggests a continuous recruitment rate and continuous mortality throughout the age classes, a closer look at individual tree stands shows that they differ strongly from each other. This suggests discontinuous recruitment patterns, which is consistent with the well-known fact that stand-replacing fires regularly rejuvenate forests in the permafrost ecosystems of our research area (Kharuk et al. 2011).

## 4.4 Allometries

We see that the tree species have very different allometries. This may be partially due to the fact that they are actually different, and partially due to random effects of the sites, and the small sample sizes. There is little literature with which to compare our results, because commonly the diameter is used to predict height, and not the other way around (e.g. (Alexander et al. 2012; Delcourt & Veraverbeke 2022). We still chose to use height as the principal variable, as it is very easy to estimate in sparse forest stands. Nevertheless, using height as a predictor, Kajimoto et al. (1999) find a similar exponent for *Larix gmelinii* stem
weight as we found for volume.

## 4.5 Comparison of inventory and remote sensing

We find for the examined remote sensing products that predicting forest statistics on the plot base results in large errors. There are various factors that can lead to such a mismatch, as discussed by Houghton et al. 2007. Imprecision in the field measurements or the data processing may play a role (Picard et al. 2015). But likely another relevant factor is the coarse
resolution of the remote sensing data, alongside the heterogeneity of the landscape on the scale between plot size and pixel size. The Simard et al. (2011) canopy height product, for example, has a resolution of 1 km$^2$, which is more than a thousand times





our average vegetation plot size. Therefore, it cannot capture differences in canopy height below the km-scale, even though many landscape elements are smaller than this. This mismatch in resolution becomes especially relevant in the forest tundra, where the sparsity of the stands makes them difficult to detect in satellite images (Ranson et al. 2004; Montesano et al. 2016).

Another issue may be the lack of calibration of the remote sensing datasets, especially in the poorly researched area of north-eastern Siberia. Zhang et al. (2019), who investigated numerous remote sensing based forest data sets suggest that most of them suffer from a lack of validation and ground-truthing. Furthermore, Yang and Kondoh (2020) investigated the Simard et al. (2011) data set and they find that it generally overestimates small canopy heights and underestimates large ones. Santoro et al. (2018) note that the relative AGB standard deviation in eastern Siberia is among the highest in the world, indicating a large

uncertainty for this region.

A different source of error is the temporal mismatch between the acquisition of the inventory data and the remote sensing images. This varies throughout our data set, as the expeditions span a time range of 10 years, which is not accounted for in the comparisons except for the comparison with the forest loss data sets. However, in the time ranges considered here, we can assume that the differences in variables such as stand height and growing stock volume are small, due to the very low growth

rates of the forests in the region (Kajimoto et al. 2010). Only disturbances, such as wildfires and insect pests, could create large changes in growing stock in a relatively short time.

We expect that all forest loss in our area is due to fire, as we did not find any signs of deforestation due to human activities on any of the surveyed sites. While the analysis of the forest loss dataset led to the expected result that the sites with recent forest loss tend to have lower fractions of living basal area, it is still surprising that we saw some plots that were supposedly

affected by forest loss, and thus by fire, with a large part of the stand alive, both in absolute and relative terms. This may be because many forest fires in Siberia are low-intensity fires (Ponomarev et al. 2022), which are detected as burned forest in one year, even though a large part of the trees recovers by the following year. Revisiting some of our survey sites in the future may help to improve the understanding of this topic.

## 4.6   The influence of climate on forest metrics

We find that the climate explains many of the quantitative forest metrics, albeit to a limited extend. Forest metrics such as basal area and stem volume are positively correlated with summer temperatures and growing degree days. However, the observed correlations are quite weak, and the range of the forest metrics is large. This suggests that the forest we observed is spatially heterogeneous and depends on properties which vary on smaller spatial scales than the climate.

It is counterintuitive that the investigated forest metrics are negatively correlated with January temperature in our data set,

but it can be explained by January temperature being negatively correlated with July temperature ($R^2$ = 0.41, slope = -0.69) and length of the growing season ($R^2$ = 0.31, slope = -0.107). The sites near the Arctic Ocean have a less continental climate, meaning they tend to have both milder winters and cooler summers than the more southerly ones. Thus, we can not conclude that colder winters are favourable for forest growth; but on the other hand, they do not seem to do much harm either, as including January temperature into multilinear models does not increase the predictive value (adj. $R^2$) much - at most by 0.057

in the case of stem volume per ha.



There is scarcely any correlation between our observed forest metrics and precipitation, which suggests that water availability is not a limiting factor for forest growth in north-eastern Siberia. Sugimoto et al. (2002) support this hypothesis by pointing out that larch forests in these regions have a good supply of water from snowmelt, rain, or thawing permafrost, depending on the weather in any given year. Opposed to this, Kharuk et al. 2019, who investigated a larch forest on the Central Siberian

Plateau, report that since the 1990s, growth has been diminished by drought stress and extreme events, which are increasing under climate warming (e.g. 2021 extreme heat in Siberia).

### 4.7 Outlook

The analyses performed in this study do not exhaust the possibilities offered by this data set and serve purely to present the data. The fact that individual trees were measured, and related to the inventory plots, make it a very versatile data set. Some

variables that were taken in the inventory can be analysed further. Especially crown diameters and crown base have not been particularly assessed as yet. The forest inventory could be related to other data collections from the same expeditions, such as projective crown cover estimations, ground vegetation surveys, soil profiles, genetic samples, stem increment cores and stem discs. These additional samples were not collected for all individuals, but they could at least be related to a portion of the forest inventory data. Also, for some of the more recent expeditions, drone-based photogrammetric and LiDAR point clouds

exist (e.g. SiDroForest), and could provide insight into the heterogeneity of the landscape and bridge the gap between survey plot size and pixel size of satellite-derived data. Further, these cm-resolution point clouds are capable of capturing single-tree measurements and bringing them to the landscape level. A different way to fill this gap, and improving the predictions of the state of remote forests is with remote sensing products at higher resolution, such as the Boreal Forest Canopy Height data set in connection with Potapov et al. (2020), which is expected to be released soon.

Our data set can also be used to calibrate and improve current and future remote sensing products. For this purpose, researchers can rely on the individual tree measurements such as height, as well as on metrics aggregated at the plot level. The data set can serve to calculate or improve allometries for the investigated taxa, especially the two aastern Siberian larch species *Larix cajanderi* and *Larix gmelinii*.

### 5 Conclusions

We presented and analysed a data set resulting from forest inventories in various regions of north-east Siberia. A subset of the entries includes diameter measurements as well as height measurements, whereas the majority only includes height. Therefore, we computed diameter-height allometries, which are reasonably accurate overall, but show a bias for some sites. It proved difficult to predict forest metrics at the plot level, for example stem volume and basal area, from a selection of remote sensing products, as these were not strongly correlated. Among the climatic variables taken from the WorldClim data set, mean July

temperature is one of the best predictors, along with GDD0 and length of growing season, while mean January temperature and precipitation proved almost insignificant. The GlobBiomass dataset and the Simard et al. (2011) forest height product are correlated with the volume and height measurements on the survey plots, but unsuitable for predicting the latter on a small



scale. The data sets used for forest age and disturbance, often differ both from each other and the observations made in the field.

This leads us to conclude that even in our times of widely available global remote sensing data sets, field measurements like the ones presented here are still vital for the understanding of remote ecosystems such as the larch-dominated forests of northeast Siberia.

## 6   Data availability

The data are available at https://doi.pangaea.de/10.1594/PANGAEA.943547 (Miesner et al. 2022).
While still under review, the data can temporarily be accessed with the link
https://www.pangaea.de/tok/45fd6ddb6a15ac79a71d0bf9a8e5bc492dda507a.

*Acknowledgements.*  The study was supported by ERC consolidator grant no. 772852.

Parts of the fieldwork were funded by the Russian Foundation for Basic Research (Grant No. 20-35-90081) and Ministry of Science and Higher Education of Russia (Grant No. FSRG-2020-0019).

Many people from the logistic and scientific staff at Afred-Wegner-Institute (AWI) and Nort-Eastern Federal University Yakutsk (NEFU) contributed to the success of the expeditions.



## Appendix A: Overview over all vegetation plots

| Site | Expedition | Latitude (°N) | Longitude (°E) | Area (m²) | Number of trees | Most frequent species group |
|------|-----------|---------------|----------------|-----------|-----------------|----------------------------|
| 11-CH-02II | 2011_Khatanga | 71.83993 | 102.88387 | 400 | 88 | Larix |
| 11-CH-02III | 2011_Khatanga | 71.84179 | 102.87589 | 400 | 93 | Larix |
| 11-CH-06I | 2011_Khatanga | 70.66915 | 97.7121 | 400 | 31 | Larix |
| 11-CH-06III | 2011_Khatanga | 70.66498 | 97.7064 | 400 | 59 | Larix |
| 11-CH-12I | 2011_Khatanga | 72.3938 | 102.30144 | 2800 | 99 | Larix |
| 11-CH-12II | 2011_Khatanga | 72.40009 | 102.28725 | 9900 | 300 | Larix |
| 11-CH-17I | 2011_Khatanga | 72.24235 | 102.24565 | 480 | 101 | Larix |
| 11-CH-17II | 2011_Khatanga | 72.24144 | 102.22661 | 400 | 67 | Larix |
| 12-KO-02/I | 2012_Kytalyk_Kolyma | 68.38916 | 161.466171 | 400 | 219 | Larix |
| 12-KO-02/II | 2012_Kytalyk_Kolyma | 68.389936 | 161.448985 | 280 | 122 | Larix |
| 12-KO-03/I | 2012_Kytalyk_Kolyma | 68.516169 | 161.18194 | 320 | 258 | Larix |
| 12-KO-03/II | 2012_Kytalyk_Kolyma | 68.513173 | 161.195505 | 256 | 174 | Larix |
| 12-KO-04/I | 2012_Kytalyk_Kolyma | 69.051323 | 161.206493 | 400 | 118 | Larix |
| 12-KO-04/II | 2012_Kytalyk_Kolyma | 69.05362 | 161.205179 | 520 | 62 | Larix |
| 12KO05 | 2012_Kytalyk_Kolyma | 69.11836 | 161.02342 | NA | 0 | NA |
| 13-TY-02-VI | 2013_Taymyr | 72.54772 | 105.7316 | 33023.36382 | 141 | Larix |
| 13-TY-02-VII | 2013_Taymyr | 72.54884 | 105.74576 | 7156.530866 | 88 | Larix |
| 13-TY-04VI | 2013_Taymyr | 72.40887 | 105.44804 | 400 | 66 | Larix |
| 13-TY-04VII | 2013_Taymyr | 72.40401 | 105.45187 | 400 | 92 | Larix |
| 13-TY-07VI | 2013_Taymyr | 71.10012 | 100.81295 | 576 | 106 | Larix |
| 13-TY-07VII | 2013_Taymyr | 71.10598 | 100.8463 | 400 | 91 | Larix |
| 13-TY-09VI | 2013_Taymyr | 72.15067 | 102.09771 | 576 | 173 | Larix |
| 13-TY-09VII | 2013_Taymyr | 72.14365 | 102.06259 | 576 | 183 | Larix |
| 14-OM-02-V1 | 2014_Omoloy | 70.74418 | 132.698523 | 400 | 450 | Larix |
| 14-OM-02-V2 | 2014_Omoloy | 70.72644 | 132.658169 | 400 | 143 | Larix |
| 14-OM-11-V3 | 2014_Omoloy | 70.957883 | 132.570074 | 400 | 0 | NA |
| 14-OM-20-V4 | 2014_Omoloy | 70.526707 | 132.914259 | 400 | 292 | Larix |
| 14-OM-TRANS1 | 2014_Omoloy | 70.943542 | 132.777408 | 314.1592654 | 24 | Larix |
| 14-OM-TRANS2 | 2014_Omoloy | 70.939004 | 132.790487 | 314.1592654 | 25 | Larix |
| 14-OM-TRANS3 | 2014_Omoloy | 70.935714 | 132.820357 | 314.1592654 | 25 | Larix |
| 14-OM-TRANS4 | 2014_Omoloy | 70.93332 | 132.854538 | 314.1592654 | 23 | Larix |
| 14-OM-TRANS5 | 2014_Omoloy | 70.935817 | 132.868951 | 314.1592654 | 24 | Larix |
| 14-OM-TRANS6 | 2014_Omoloy | 70.944295 | 132.8777 | 314.1592654 | 22 | Larix |
| 14-OM-TRANS6-7 | 2014_Omoloy | 70.948754 | 132.884332 | NA | 0 | Larix |
| 16-KP-V01 | 2016_Keperveem | 67.3618 | 168.2542 | 706.8583471 | 37 | Larix |
| 16-KP-V02 | 2016_Keperveem | 67.366 | 168.2366 | 706.8583471 | 7 | Larix |
| 16-KP-V03 | 2016_Keperveem | 67.3664 | 168.2948 | 624 | 128 | Larix |
| 16-KP-V04 | 2016_Keperveem | 67.3736 | 168.31 | 706.8583471 | 13 | Larix |
| 16-KP-V05 | 2016_Keperveem | 67.3769 | 168.3122 | 706.8583471 | 107 | Larix |
| 16-KP-V06 | 2016_Keperveem | 67.35 | 168.1885 | 706.8583471 | 107 | Larix |
| 16-KP-V07 | 2016_Keperveem | 67.3456 | 168.1842 | 706.8583471 | 0 | Larix |
| 16-KP-V08 | 2016_Keperveem | 67.3449 | 168.1802 | 706.8583471 | 1 | Larix |
| 16-KP-V09 | 2016_Keperveem | 67.3538 | 168.2157 | 706.8583471 | 0 | Larix |
| 16-KP-V10 | 2016_Keperveem | 67.3452 | 168.2013 | 706.8583471 | 24 | Larix |

**Table A1 (1/5).** Overview over all vegetation plots





| 16-KP-V11 | 2016_Keperveem | 67.35 | 168.2009 | 706.8583471 | 85 | Larix |
|---|---|---|---|---|---|---|
| 16-KP-V12 | 2016_Keperveem | 67.3531 | 168.2264 | 706.8583471 | 68 | Larix |
| 16-KP-V13 | 2016_Keperveem | 66.9731 | 163.4177 | 706.8583471 | 187 | Larix |
| 16-KP-V14 | 2016_Keperveem | 66.9874 | 163.3981 | 706.8583471 | 14 | Larix |
| 16-KP-V15 | 2016_Keperveem | 66.9914 | 163.3843 | 706.8583471 | 1 | Larix |
| 16-KP-V16 | 2016_Keperveem | 66.9715 | 163.4021 | 706.8583471 | 31 | Larix |
| 16-KP-V17 | 2016_Keperveem | 66.9869 | 163.455 | 480 | 190 | Larix |
| 16-KP-V18 | 2016_Keperveem | 66.9699 | 163.3845 | 50 | 192 | Larix |
| 16-KP-V19 | 2016_Keperveem | 66.9706 | 163.3948 | 706.8583471 | 238 | Larix |
| 16-KP-V20 | 2016_Keperveem | 65.9249 | 166.3609 | 706.8583471 | 107 | Larix |
| 16-KP-V21 | 2016_Keperveem | 65.926 | 166.3609 | 706.8583471 | 48 | Larix |
| 16-KP-V22 | 2016_Keperveem | 65.9352 | 166.3905 | 706.8583471 | 6 | Larix |
| 16-KP-V23 | 2016_Keperveem | 65.9352 | 166.3933 | 706.8583471 | 0 | Larix |
| 16-KP-V24 | 2016_Keperveem | 65.9365 | 166.389 | 706.8583471 | 0 | Larix |
| 16-KP-V25 | 2016_Keperveem | 65.9372 | 166.3906 | 706.8583471 | 0 | Larix |
| 16-KP-V26 | 2016_Keperveem | 65.9369 | 166.3861 | 706.8583471 | 76 | Larix |
| 16-KP-V27 | 2016_Keperveem | 65.9369 | 166.385 | 706.8583471 | 114 | Larix |
| 16-KP-V28 | 2016_Keperveem | 65.9231 | 166.3683 | 1296 | 96 | Larix |
| 16-KP-V29 | 2016_Keperveem | 65.9252 | 166.3882 | 706.8583471 | 49 | Larix |
| 16-KP-V30 | 2016_Keperveem | 65.9579 | 166.3333 | 706.8583471 | 4 | Larix |
| 16-KP-V31 | 2016_Keperveem | 65.9585 | 166.3368 | 706.8583471 | 0 | Larix |
| 16-KP-V32 | 2016_Keperveem | 65.9468 | 166.3561 | 706.8583471 | 6 | Larix |
| 16-KP-V33 | 2016_Keperveem | 65.9459 | 166.3577 | 706.8583471 | 0 | Larix |
| 16-KP-V34 | 2016_Keperveem | 65.9415 | 166.3486 | 706.8583471 | 140 | Larix |
| 16-KP-V35 | 2016_Keperveem | 65.9329 | 166.2618 | 706.8583471 | 125 | Larix |
| 16-KP-V36 | 2016_Keperveem | 65.9294 | 166.291 | 706.8583471 | 2 | Larix |
| 16-KP-V37 | 2016_Keperveem | 65.9002 | 166.419 | 576 | 90 | Larix |
| 16-KP-V38 | 2016_Keperveem | 65.9003 | 166.4168 | 706.8583471 | 135 | Larix |
| 16-KP-V39 | 2016_Keperveem | 65.9217 | 166.3139 | 706.8583471 | 205 | Larix |
| 16-KP-V40 | 2016_Keperveem | 67.7969 | 168.7096 | 706.8583471 | 0 | NA |
| 16-KP-V41 | 2016_Keperveem | 67.8171 | 168.6865 | 706.8583471 | 0 | NA |
| 16-KP-V42 | 2016_Keperveem | 67.8171 | 168.6885 | 706.8583471 | 0 | NA |
| 16-KP-V43 | 2016_Keperveem | 67.8195 | 168.6976 | 706.8583471 | 0 | NA |
| 16-KP-V44 | 2016_Keperveem | 67.8196 | 168.6963 | 706.8583471 | 0 | NA |
| 16-KP-V45 | 2016_Keperveem | 67.82 | 168.714 | 706.8583471 | 0 | NA |
| 16-KP-V46 | 2016_Keperveem | 67.8199 | 168.7115 | 706.8583471 | 0 | NA |
| 16-KP-V47 | 2016_Keperveem | 67.8048 | 168.7037 | 706.8583471 | 0 | NA |
| 16-KP-V48 | 2016_Keperveem | 67.8002 | 168.6379 | 706.8583471 | 0 | NA |
| 16-KP-V49 | 2016_Keperveem | 67.8026 | 168.6359 | 706.8583471 | 0 | NA |
| 16-KP-V50 | 2016_Keperveem | 67.8051 | 168.6297 | 706.8583471 | 0 | NA |
| 16-KP-V51 | 2016_Keperveem | 67.8055 | 168.6327 | 706.8583471 | 0 | NA |
| 16-KP-V52 | 2016_Keperveem | 67.8069 | 168.6311 | 706.8583471 | 0 | NA |
| 16-KP-V53 | 2016_Keperveem | 67.8079 | 168.6323 | 706.8583471 | 0 | NA |
| 16-KP-V54 | 2016_Keperveem | 67.8096 | 168.6299 | 706.8583471 | 0 | NA |
| 16-KP-V55 | 2016_Keperveem | 67.8091 | 168.6336 | 706.8583471 | 0 | NA |
| 16-KP-V56 | 2016_Keperveem | 67.8082 | 168.6355 | 706.8583471 | 0 | NA |


**Table A1 (2/5).** Overview over all vegetation plots





| 16-KP-V57 | 2016_Keperveem | 67.8076 | 168.645 | 706.8583471 | 0 | NA |
|---|---|---|---|---|---|---|
| 16-KP-V58 | 2016_Keperveem | 67.8086 | 168.645 | 706.8583471 | 0 | NA |
| 18-LD-VP012-Tit-Ary | 2018_Lena | 71.967274 | 127.092825 | 900 | 0 | Larix |
| B19-T1 | 2019_Batagay | 67.58117 | 134.785314 | 706.8583471 | 0 | NA |
| B19-T2 | 2019_Batagay | 67.580618 | 134.78351 | 706.8583471 | 0 | NA |
| EN18000 | 2018_Chukotka | 68.097147 | 166.375447 | 706.8583471 | 111 | Larix |
| EN18001 | 2018_Chukotka | 67.39273 | 168.34662 | 706.8583471 | 50 | Larix |
| EN18002 | 2018_Chukotka | 67.386775 | 168.336731 | 706.8583471 | 0 | NA |
| EN18003 | 2018_Chukotka | 67.39691 | 168.34702 | 706.8583471 | 37 | Larix |
| EN18004 | 2018_Chukotka | 67.397489 | 168.351225 | 706.8583471 | 6 | Larix |
| EN18005 | 2018_Chukotka | 67.419652 | 168.387511 | 706.8583471 | 1 | Larix |
| EN18006 | 2018_Chukotka | 67.414969 | 168.402874 | 706.8583471 | 141 | Larix |
| EN18007 | 2018_Chukotka | 67.403274 | 168.371965 | 706.8583471 | 181 | Larix |
| EN18008 | 2018_Chukotka | 67.402135 | 168.375284 | 706.8583471 | 0 | Larix |
| EN18009 | 2018_Chukotka | 67.400725 | 168.379683 | 706.8583471 | 4 | Larix |
| EN18010 | 2018_Chukotka | 67.402371 | 168.3662 | 706.8583471 | 11 | Larix |
| EN18011 | 2018_Chukotka | 67.404042 | 168.364252 | 706.8583471 | 0 | Salix |
| EN18012 | 2018_Chukotka | 67.402142 | 168.378078 | 706.8583471 | 80 | Larix |
| EN18013 | 2018_Chukotka | 67.405174 | 168.355304 | 706.8583471 | 0 | Salix |
| EN18014 | 2018_Chukotka | 67.395309 | 168.349106 | 1600 | 59 | Larix |
| EN18015 | 2018_Chukotka | 67.420379 | 168.33061 | 706.8583471 | 0 | Salix |
| EN18016 | 2018_Chukotka | 67.426726 | 168.390047 | 706.8583471 | 0 | Larix |
| EN18017 | 2018_Chukotka | 67.43229 | 168.383376 | 706.8583471 | 0 | Salix |
| EN18018 | 2018_Chukotka | 67.456295 | 168.405961 | 706.8583471 | 0 | NA |
| EN18019 | 2018_Chukotka | 67.457073 | 168.408963 | 706.8583471 | 0 | NA |
| EN18020 | 2018_Chukotka | 67.459159 | 168.411934 | 706.8583471 | 0 | NA |
| EN18021 | 2018_Chukotka | 67.392129 | 168.328815 | 706.8583471 | 116 | Larix |
| EN18022 | 2018_Chukotka | 67.401024 | 168.348006 | 706.8583471 | 0 | Larix |
| EN18023 | 2018_Chukotka | 67.399236 | 168.351285 | 706.8583471 | 0 | Pinus pumila |
| EN18024 | 2018_Chukotka | 67.370964 | 168.426362 | 706.8583471 | 120 | Larix |
| EN18025 | 2018_Chukotka | 67.367027 | 168.42381 | 706.8583471 | 97 | Larix |
| EN18026 | 2018_Chukotka | 67.396089 | 168.354297 | 706.8583471 | 77 | Larix |
| EN18027 | 2018_Chukotka | 67.393408 | 168.35905 | 706.8583471 | 54 | Larix |
| EN18028 | 2018_Chukotka | 68.46781 | 163.357622 | 706.8583471 | 97 | Larix |
| EN18029 | 2018_Chukotka | 68.465606 | 163.352262 | 706.8583471 | 71 | Larix |
| EN18030 | 2018_Chukotka | 68.405539 | 164.532731 | 706.8583471 | 669 | Larix |
| EN18031 | 2018_Chukotka | 68.404918 | 164.545351 | 706.8583471 | 100 | Larix |
| EN18032 | 2018_Chukotka | 68.404868 | 164.551181 | 706.8583471 | 1 | Salix |
| EN18033 | 2018_Chukotka | 68.403212 | 164.551805 | 706.8583471 | 0 | Salix |
| EN18034 | 2018_Chukotka | 68.403486 | 164.548043 | 706.8583471 | 35 | Larix |
| EN18035 | 2018_Chukotka | 68.403166 | 164.590932 | 706.8583471 | 168 | Larix |
| EN18051 | 2018_Chukotka | 67.80261 | 168.7047 | 706.8583471 | 0 | NA |
| EN18052 | 2018_Chukotka | 67.79941 | 168.7083 | 706.8583471 | 0 | NA |
| EN18053 | 2018_Chukotka | 67.79729 | 168.7107 | 706.8583471 | 0 | NA |
| EN18054 | 2018_Chukotka | 67.79766 | 168.6904 | 706.8583471 | 0 | NA |
| EN18055 | 2018_Chukotka | 67.79103 | 168.6825 | 706.8583471 | 0 | NA |

**Table A1 (3/5).** Overview over all vegetation plots





| EN18061 | 2018_Yakutia | 62.076376 | 129.618586 | 706.8583471 | 611 | Pinus tree |
| EN18062 | 2018_Yakutia | 62.179065 | 127.805796 | 706.8583471 | 418 | Larix |
| EN18063 | 2018_Yakutia | 63.776636 | 122.501003 | 706.8583471 | 459 | Larix |
| EN18064 | 2018_Yakutia | 63.814594 | 122.209683 | 706.8583471 | 435 | Pinus tree |
| EN18065 | 2018_Yakutia | 63.795223 | 122.443715 | 304 | 242 | Larix |
| EN18066 | 2018_Yakutia | 63.797119 | 122.438071 | 706.8583471 | 115 | Larix |
| EN18067 | 2018_Yakutia | 63.076368 | 117.975342 | 706.8583471 | 339 | Larix |
| EN18068 | 2018_Yakutia | 63.074232 | 117.98207 | 706.8583471 | 74 | Larix |
| EN18069 | 2018_Yakutia | 63.173288 | 118.132507 | 706.8583471 | 543 | Picea |
| EN18070_centre | 2018_Yakutia | 63.082476 | 117.985333 | 300 | 81 | Picea |
| EN18070_edge | 2018_Yakutia | 63.082983 | 117.984938 | 300 | 224 | Picea |
| EN18070_end | 2018_Yakutia | 63.08341 | 117.984574 | 200 | 0 | NA |
| EN18070_transition | 2018_Yakutia | 63.082733 | 117.985156 | 300 | 142 | Picea |
| EN18071 | 2018_Yakutia | 62.225093 | 116.275603 | 706.8583471 | 236 | Larix |
| EN18072 | 2018_Yakutia | 62.199571 | 117.379125 | 706.8583471 | 688 | Larix |
| EN18073 | 2018_Yakutia | 62.188712 | 117.409917 | 706.8583471 | 837 | Larix |
| EN18074 | 2018_Yakutia | 62.215192 | 117.021599 | 706.8583471 | 275 | Picea |
| EN18075 | 2018_Yakutia | 62.696991 | 113.676535 | 706.8583471 | 274 | Pinus tree |
| EN18076 | 2018_Yakutia | 62.70089 | 113.67341 | 706.8583471 | 582 | Larix |
| EN18077 | 2018_Yakutia | 61.892568 | 114.288623 | 706.8583471 | 546 | Pinus tree |
| EN18078 | 2018_Yakutia | 61.575058 | 114.29995 | 706.8583471 | 236 | Larix |
| EN18079 | 2018_Yakutia | 59.974919 | 112.958985 | 706.8583471 | 305 | Pinus tree |
| EN18080 | 2018_Yakutia | 59.977106 | 112.961379 | 706.8583471 | 339 | Picea |
| EN18081 | 2018_Yakutia | 59.970583 | 112.987096 | 706.8583471 | 83 | Picea |
| EN18082 | 2018_Yakutia | 59.97764 | 112.98218 | 706.8583471 | 101 | Larix |
| EN18083 | 2018_Yakutia | 59.974714 | 113.002874 | 706.8583471 | 138 | Picea |
| EN21-201 | 2021_Yakutia | 63.217776 | 139.543709 | NA | 0 | Larix |
| EN21-202 | 2021_Yakutia | 63.32516 | 141.07455 | 706.8583471 | 160 | Larix |
| EN21-203 | 2021_Yakutia | 63.430107 | 140.412509 | 706.8583471 | 126 | Larix |
| EN21-204 | 2021_Yakutia | 63.44253 | 140.40282 | 706.8583471 | 118 | Larix |
| EN21-205 | 2021_Yakutia | 63.43858 | 140.40688 | 706.8583471 | 44 | Larix |
| EN21-206 | 2021_Yakutia | 63.34379 | 141.07071 | 706.8583471 | 81 | Larix |
| EN21-207 | 2021_Yakutia | 63.344383 | 141.069788 | NA | 3 | Pinus pumila |
| EN21-208 | 2021_Yakutia | 63.34528 | 141.06827 | NA | 0 | NA |
| EN21-209 | 2021_Yakutia | 63.39854 | 140.55406 | 706.8583471 | 50 | Larix |
| EN21-210 | 2021_Yakutia | 63.397717 | 140.55925 | NA | 0 | NA |
| EN21-211 | 2021_Yakutia | 63.40056 | 140.55357 | 706.8583471 | 109 | Larix |
| EN21-212 | 2021_Yakutia | 63.232626 | 142.962381 | 706.8583471 | 251 | Larix |
| EN21-213 | 2021_Yakutia | 63.230378 | 142.963774 | 100 | 219 | Larix |
| EN21-214 | 2021_Yakutia | 63.23257 | 142.9577 | NA | 0 | NA |
| EN21-215 | 2021_Yakutia | 63.210719 | 139.540937 | 706.8583471 | 14 | Larix |
| EN21-216 | 2021_Yakutia | 63.212267 | 139.541692 | NA | 0 | NA |
| EN21-217 | 2021_Yakutia | 63.438697 | 140.597609 | 706.8583471 | 41 | Larix |
| EN21-218 | 2021_Yakutia | 63.428277 | 140.579547 | 706.8583471 | 0 | NA |
| EN21-219 | 2021_Yakutia | 63.425647 | 140.588331 | 706.8583471 | 284 | Larix |
| EN21-220 | 2021_Yakutia | 62.07984 | 132.3668 | NA | 0 | NA |

**Table A1 (4/5).** Overview over all vegetation plots





| EN21-221 | 2021_Yakutia | 62.083241 | 132.372643 | 706.8583471 | 28 | Betula |
|----------|--------------|-----------|------------|-------------|------|--------|
| EN21-222 | 2021_Yakutia | 62.08595 | 132.370772 | 706.8583471 | 640 | Larix |
| EN21-223 | 2021_Yakutia | 62.087193 | 132.370561 | 706.8583471 | 306 | Larix |
| EN21-224 | 2021_Yakutia | 62.042778 | 132.388521 | 706.8583471 | 0 | NA |
| EN21-225 | 2021_Yakutia | 62.044236 | 132.391202 | 706.8583471 | 452 | Betula |
| EN21-226 | 2021_Yakutia | 62.045558 | 132.389098 | 706.8583471 | 168 | Larix |
| EN21-227 | 2021_Yakutia | 62.040546 | 132.396302 | 314.15926535897 9 | 4 | Larix |
| EN21-228 | 2021_Yakutia | 62.384988 | 133.748979 | 706.8583471 | 268 | Larix |
| EN21-229 | 2021_Yakutia | 62.384468 | 133.750727 | 314.1592654 | 109 | Larix |
| EN21-230 | 2021_Yakutia | 62.334507 | 133.688018 | 706.8583471 | 163 | Larix |
| EN21-231 | 2021_Yakutia | 62.334694 | 133.68405 | NA | 22 | Betula |
| EN21-232 | 2021_Yakutia | 62.172203 | 130.911195 | 706.8583471 | 652 | Larix |
| EN21-233 | 2021_Yakutia | 62.169607 | 130.903851 | 706.8583471 | 308 | Larix |
| EN21-234 | 2021_Yakutia | 62.287013 | 130.377589 | 706.8583471 | 39 | Pinus tree |
| EN21-235 | 2021_Yakutia | 62.275634 | 130.37659 | 706.8583471 | 141 | Pinus tree |
| EN21-236 | 2021_Yakutia | 62.262231 | 130.327876 | 706.8583471 | 234 | Pinus tree |
| EN21-237 | 2021_Yakutia | 62.13009 | 130.874837 | 706.8583471 | 288 | Larix |
| EN21-238 | 2021_Yakutia | 62.133528 | 130.873521 | 706.8583471 | 176 | Larix |
| EN21-239 | 2021_Yakutia | 62.316127 | 130.116028 | 314.1592654 | 290 | Alnus |
| EN21-240 | 2021_Yakutia | 62.353399 | 130.151416 | 706.8583471 | 645 | Picea |
| EN21-241 | 2021_Yakutia | 62.148377 | 130.65177 | 706.8583471 | 29 | Larix |
| EN21-242 | 2021_Yakutia | 62.148415 | 130.653568 | 706.8583471 | 445 | Betula |
| EN21-243 | 2021_Yakutia | 62.149423 | 130.654024 | 706.8583471 | 0 | NA |
| EN21-244 | 2021_Yakutia | 62.156934 | 130.659589 | 314.1592654 | 628 | Larix |
| EN21-245 | 2021_Yakutia | 61.78444 | 130.48492 | 706.8583471 | 299 | Populus |
| EN21-246 | 2021_Yakutia | 61.78305 | 130.49245 | 225 | 2713 | Betula |
| EN21-247 | 2021_Yakutia | 61.77975 | 130.49998 | 706.8583471 | 76 | Larix |
| EN21-248 | 2021_Yakutia | 61.747877 | 130.530323 | 706.8583471 | 405 | Betula |
| EN21-249 | 2021_Yakutia | 61.745655 | 130.530715 | 706.8583471 | 835 | Larix |
| EN21-250 | 2021_Yakutia | 61.745696 | 130.532625 | 706.8583471 | 539 | Betula |
| EN21-251 | 2021_Yakutia | 61.740083 | 130.528577 | 706.8583471 | 149 | Larix |
| EN21-252 | 2021_Yakutia | 61.897154 | 130.482395 | 706.8583471 | 352 | Salix |
| EN21-253 | 2021_Yakutia | 61.89501 | 130.4848 | 706.8583471 | 290 | Larix |
| EN21-254 | 2021_Yakutia | 61.894779 | 130.488766 | 706.8583471 | 291 | Larix |
| EN21-255 | 2021_Yakutia | 61.769113 | 130.386747 | 706.8583471 | 871 | Larix |
| EN21-256 | 2021_Yakutia | 61.76639 | 130.83875 | 706.8583471 | 596 | Betula |
| EN21-257 | 2021_Yakutia | 61.770502 | 130.391538 | NA | 0 | NA |
| EN21-258 | 2021_Yakutia | 61.899226 | 130.423401 | 706.8583471 | 506 | Larix |
| EN21-259 | 2021_Yakutia | 61.901329 | 130.500516 | 706.8583471 | 492 | Larix |
| EN21-260 | 2021_Yakutia | 61.76387 | 130.47968 | 706.8583471 | 309 | Larix |
| EN21-261 | 2021_Yakutia | 61.766817 | 130.457716 | 706.8583471 | 329 | Larix |
| EN21-262 | 2021_Yakutia | 61.76123 | 130.47043 | NA | 0 | NA |
| EN21-263 | 2021_Yakutia | 62.209135 | 127.691498 | NA | 0 | NA |
| EN21-264 | 2021_Yakutia | 62.216896 | 127.717821 | NA | 0 | NA |

**Table A1 (5/5).** Overview over all vegetation plots



## Appendix B: Coefficients of Diameter-Height-Allometries

Allometries were calculated, to obtain the diameter from the height of the tree, with the formula

$$D = (a_1 + a_3 \cdot S) \cdot H^{(a_2 + a_4 \cdot S)}$$

where $D$ is the diameter in cm, $H$ is the height, and $S$ is the stand density, obtained from the number of trees per hectare ($T_{ha}$), as follows:

$$S = \max(\log_{10}(T_{ha}), 2))$$

The coefficients $a_1$, $a_2$, $a_3$ and $a_4$ resulted from fitting with the least squares method are shown in tables B1 and B2.

**Diameter at Base**

| Species group | a1 | a2 | a3 | a4 | Standard error |
|---|---|---|---|---|---|
| Larix | 4.42635585352109 | -0.776796173348 | 0.769611594482003 | 0.0784301866347613 | 4.90874601139169 |
| Salix | 2.44808930170272 | 0 | 0.785304295538417 | 0 | 4.24060655519071 |
| Betula | 0.912483091103982 | 0 | 1.15137420489562 | 0 | 4.32279975856903 |
| Alnus | 0.942870833892095 | 0 | 0.948313878522045 | 0 | 2.3793245844191 |
| Pinus tree | 1.86443570083814 | 0 | 1.02819641344765 | 0 | 4.5821377938073 |
| Picea | 0.917765224118778 | 0 | 1.1498526562509 | 0 | 4.08174860543055 |
| unclassyfied | 3.45480540316162 | 0 | 0.597988384981508 | 0 | 3.18450507223829 |
| Abies | 2.6910468742368 | 0 | 0.662455832295806 | 0 | 1.00147231307059 |
| Populus | 0.0576372543717623 | 0 | 2.28587320664546 | 0 | 2.90143999743121 |
| Larix krumholz | 3.92779305391239 | 0 | 0.580696141176095 | 0 | 2.17629401965687 |

**Table B1.** Coefficients for diameter at base alllometries

## Appendix C: Correlation coefficients for forest canopy height

In section 3.2.3, linear correlations were calculated between the Simard et al. forest height product and different forest metrics (heights in m), with the results shown in table C1.

*Author contributions.* TM and UH conceptualized the manuscript and the analysis. TM drafted the manuscript and performed the analysis. SK revised the data. All authors participated in field work and revised the manuscript.

*Competing interests.* No competing interests are declared.



**Diameter at Breast Height**

| Species group | a1 | a2 | a3 | a4 | Standard error |
|---|---|---|---|---|---|
| Larix | 5.55117505398728 | -1.052830857608 | 0.47568985561217 | 0.124755584501273 | 3.96003451855488 |
| Salix | 2.10313653768383 | 0 | 0.701087578816537 | 0 | 1.72917862876573 |
| Betula | 0.840246310768036 | 0 | 1.07641382238372 | 0 | 3.06235268623208 |
| Alnus | 1.9434531823507 | 0 | 0.479914784240546 | 0 | 2.57138482210808 |
| Pinus tree | 1.87747899970649 | 0 | 0.967151587180109 | 0 | 4.20737101659919 |
| Picea | 1.19468336091674 | 0 | 0.98264915187966 | 0 | 3.06105794788814 |
| unclassyfied | 3.0453919739044 | 0 | 0.585378582934893 | 0 | 2.76608980661583 |
| Abies | 2.56592075450757 | 0 | 0.627480357100185 | 0 | 0.547445005612979 |
| Populus | 0.143953004791932 | 0 | 1.89866259734449 | 0 | 2.51911067866835 |
| Larix krumholz | 2.68253262219705 | 0 | 0.766163020810581 | 0 | 1.12855406216603 |

**Table B2.** Coefficients for diameter at breast height allometries

| Variable | adj. $R^2$ | std. error |
|---|---|---|
| Maximum height | 0.190 | 5.42 |
| Height 98th percentile | 0.152 | 4.638 |
| Height 90th percentile | 0.0961 | 4.063 |
| Height 75th percentile | 0.0522 | 3.359 |
| Height 25th percentile | 0.0115 | 1.579 |
| Mean height | 0.0688 | 2.143 |

**Table C1.** Correlation coefficients for forest canopy height.

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
