# Peer review of "Forest structure and individual tree inventories of north-eastern Siberia along climatic gradients"

_Earth System Science Data, 2022_

## Referee Comment (RC1)

The manuscript *Forest structure and individual tree inventories of north-eastern Siberia along climactic gradients* by Miesner and colleagues is (1) a dataset of forest surveys in NE Siberia, (2) a brief analysis of the data set, and (3) a comparison of the field data to remote sensing products. The data are a truly important contribution to forest inventory data in NE Siberia, which are notably scarce considering the region's large extent and importance to global biogeochemical cycles. The data are reported clearly and consistently, and the methods and analyses employed by the authors are appropriate to answer their study questions.

Below, I detail my comments on (1) the dataset and (2) the manuscript.

**Dataset available on PANGEA**

In general, I found the data presentation easy to follow. The metadata are easily accessible, columns are defined, and units are easily findable. However, I suggest the following improvements:

- The data are currently organized with the metadata in the same file as the tabular data. This is problematic for reading the tabular data into R. I had to spend quite a bit of time figuring out how to delete the top metadata and then reading the tabular data into R correctly. I suggest I suggest having one file for metadata and one file for data, rather than combing them into one file.
- Within one data file, there are multiple columns with the same name (e.g., 'Plot', 'Tree height', 'Area/locality'), which is needlessly confusing. Some of the column names are very long and include spaces and parentheses (e.g., 'Height quantile [m] (Quantile (25th))'), which can be difficult for working with computer programs. I suggest that data columns be short and unique (e.g., 'Federation', 'District', etc.), or at the very least include an underscore (e.g., 'Location_federation', 'Location_District', etc.).
- Some of the comments in the database are in German; could these be translated into English? For example:

  **13-TY-09-VII** 🔍 * *Latitude:* 72.143650 * *Longitude:* 102.062590 * *Date/Time:* 2013-07-31T00:00:00 * *Campaign:* RU-Land_2013_Taymyr (Taymyr2013) 🔍 * *Basis:* AWI Arctic Land Expedition 🔍 * *Method/Device:* Sampling by hand (HAND) 🔍 * *Comment:* Sehr lichte Fläche, viele Moose und Senken mit Eriophorum sind eingestreut; wenige sehr dicke Bäume; einiges an Totholz auf der Fläche; Rentierspuren und Kot vorgefunden; viele Lemminglöcher

- The term 'Krumholz' is used in the dataset but not discussed in the text. For clarity, I suggest discussing how growth types (tree vs. krumholz) were determined.
- Were the latitude/longitude coordinates taken plot center or somewhere else? Was the WGS84 datum used to collect the coordinates? This information is important for geolocating in relationship to satellite products.

**Manuscript**

In general, I found the manuscript easy to read and easy to follow. The figures are easily understandable. That being said, the analytical methods (PCA, multilinear regression) need more explanation and in general, the Discussion could benefit from more rigorous analysis. I detail these points in the line-by-line comments below.

Line-by-line comments:

L4: Can you mention how many plots/site?

L 22: '… with much of these forests located in.."

L36-37: Do you mean there only a few inventories? What kind of 'studies' are you referring to here? There are many more papers that deal explicitly with larch dominated ecosystems of Russia, although maybe they are not inventories as defined here. The addition of a few descriptive words would suffice here.

L44-49: While it is true that these aspects are difficult to capture with remote sensing, there have been major advances in this area in the past decade and 'can hardly be captured remotely at all' does not accurately reflect recent advances.

L60: remove 'extremely'

L61: What kind of exceptions (more or less precipitation?)?

L107: Which variables? (Please list in this sentence). Also, there should be a flag in the dataset for whether the variable was measured directly or derived, which I do not see.

L141: "...observations of Landsat images (30 m resolution)."

L144 Specify that the tree cover gain/loss is binary (gained/lost) and not continuous (% lost/gained), as the wording currently implies.

L144-145 'The tree cover loss product is thus derived from the annual forest cover products…" this does not make sense to me, since there is not an annual tree cover product published by Hansen. The only annual product is loss/gain.

L154: Delete 'however'

L151: Is the value 100/no data relevant to this analysis? These specifics seem unnecessary.

L155: More information is needed about how the PCA was conducted. What package in R? Were any data transformations performed prior to ordination?

L158: More information is needed about how the multilinear regressions were performed. Were variables checked for the assumption of normality? Were variables removed for collinearity? How was the best model selected? Were the residuals checked for homogeneity of variance, etc. Why were univariate regressions performed in addition to multivariate? (I see no reason to perform a univariate regression when multiple predictor variables exist.) Which R packages was used?

L162: It is unclear to me what you are assessing with the t-test. As written, it implies you determined the sites with recent tree cover loss with a t-test, but on what? And doesn't the Hansen data show tell you that directly?

L169: Are these double inventoried trees flagged in the data?  How can we distinguish them?

L172: delete 'only' and 'while 60 do not.'

L173: Presenting these numbers as percentages rather than absolute numbers would aid the reader in understanding.

L176: change to 'the two *Larix* species never occur…'

L179: delete 'only' and 'up'

L198: It is not clear to me why these plots were singled out. Are they representative of other plots? Instead of discussing these individual plots, I suggest discussing the meaning of the average Gini coefficient and whether you saw any regional differences in the Gini coefficient. Figure 5 is not discussed at all in the text, and this would be a good place for it.

L212: Re-word to say that *Pinus pumila* and *Salix spp.* occur more frequently between 65N and 70N.

L217: Which variables were highly correlated with the first (and second) principal component(s)? As written, the PCA plays a very small role in this paper, so you could consider deleting.

L223: I suggest focusing on the variables with statistically significant relationships rather than the $R^2$, which is likely to be quite low given how noisy ecological field data can be. One idea would be to update figure 7/this section to only include significant relationships, as determined by the properly selected multilinear model. This will draw the reader to which variables are important.

L227: A multilinear model is more relevant than individual regressions. When properly implemented, you can control for collinearity and report which variables are statistically significant. An $R^2$ of 0.356 is not unreasonable.

L240: Appendix is missing an A

L242: ORNL DAAC is the hosting service. Chen et al. is the correct citation.

L242: Does the field data at these 6 sites indicate there has been a stand replacing fire over this period? (i.e., does the field data corroborate the remote sensing data)?

L243: Does the Hansen data set also show losses where there have been stand replacing fires?

L243: This wording is unclear. Do you mean that there are 5 plots where Hansen shows forest loss and Chen at al do not? What do the field data say about those plots?

L247: Regarding the plots where dead trees do not contribute a relevant amount of the basal area, was there evidence of logging at these sites? Or perhaps the dead trees are not standing but fallen? This information is helpful in determining how well the Hansen data represent what is happening on the ground.

L257: delete 'very'

270: Here, a distinction should be made between the landscape-scale distribution (all plots combined), which seems to be close-to exponential, and the stand-level distribution (looking at

plots individually), which are not. It makes sense that at the landscape-scale, we would see continuous recruitment, but at the stand-scale, recruitment may be more episodic, likely in response to fires.

L275: Is there reason to believe that different species should NOT have different allometries? I thought the expectation is that allometries are species-specific.

L275: You may be able to determine if allometries for larch or pine vary by climate. For example, Berner et al., 2015 found that in some (but not all) species, boreal shrub allometries vary by ecoregion.

L276: What are the sample sizes?

L277: It should be fairly straightforward to calculate the relationship between the derived DBHs and your measured heights in a way that is comparable with data from the literature.

L293: Is this Santoro et al. (2018) a or b?

L293: Please give some context for the Santoro et al. (2018) sentence. What data set are they working with or assessing?

L300: What about logging by local communities?

L310: Extend = extent

L315: Reporting results from the multilinear model is more relevant here. Using model selection including variance inflation factors will help you get rid of variables that are collinear.

L319: Did you preform proper model selection? It seems that January temp may be too highly correlated with July temp and/or not important enough to be included in a multiple regression model.

L326: Please include a reference for this; Dobricic et al., 2020 or Collow et al., 2022 discuss extreme events, but please also include a reference for increasing drought stress.

L332: How can the reader access this additional data?

L339: The wording implies that the Potapov data set is expected to be released soon, but the citation says it was published in 2020.

L339: Please describe what the Potapov paper is about.

L343: aastern = eastern?

Figure 2:

    -Please include the definition of DBS and DBH in the caption.

    -It is difficult to see the colors of the different species groups. Can you make the circles in the legend bigger? Perhaps also just a little bigger in the graph.

-The term 'krumholz' appears in this figure but nowhere in the text. How is it defined here?

-Are the allometries here just for larch krumholz and not for larch trees?

-For the density allometries, are the different colored lines different quartiles? It seems like the bright yellow one (highest density) is not fit to any data (there are no bright yellow points, especially as height increases). It also seems like the high density line (yellow-grey) is below the low density (dark purple) and medium density (purple grey) lines but the highest density (bright yellow) line is above all of them. Why would this be (i.e., why isn't there a negative progression from low to medium to high to highest)?

-In caption, can you clarify that the regression lines in the bottom graphs are for illustrative purposes only, because you used stand density as a continuous variable (rather than binned) to calculate DBH and DBS (or at least, this is what I understand from the methods)?

Figure 5: This figure is not discussed in the text. Also, please make the legend and entire plot larger for enhanced readability.

Figure 6: It is not clear to me what this plot adds, and it is not discussed much in the text.

Figure 8:

-Please report $R^2$ and p-values for both plots.

- Please discuss (in the text) what it means that the data fit better under the logarithmic transformation.

References

Berner, L. T., Alexander, H. D., Loranty, M. M., Ganzlin, P., Mack, M. C., Davydov, S. P., and Goetz, S. J.: Biomass allometry for alder, dwarf birch, and willow in boreal forest and tundra ecosystems of far northeastern Siberia and north-central Alaska, 337, 110–118, https://doi.org/10.1016/j.foreco.2014.10.027, 2015.

Collow, A. B. M., Thomas, N. P., Bosilovich, M. G., Lim, Y.-K., Schubert, S. D., and Koster, R. D.: Seasonal Variability in the Mechanisms Behind the 2020 Siberian Heatwaves, 1–44, https://doi.org/10.1175/JCLI-D-21-0432.1, 2022.

Dobricic, S., Russo, S., Pozzoli, L., Wilson, J., and Vignati, E.: Increasing occurrence of heat waves in the terrestrial Arctic, 15, 024022, https://doi.org/10.1088/1748-9326/ab6398, 2020.

---

## Author Comment (AC1)

We would like to express our gratitude to Anonymous reviewer 1 for the considerate revision of the paper, and for the attention to detail in formulation the numerous comments, which helped us to improve the paper.

A revised version of the manuscript has been prepared based on the reviews.

Reviewer 1 wrote:

The manuscript *Forest structure and individual tree inventories of north-eastern Siberia along climactic gradients* by Miesner and colleagues is (1) a dataset of forest surveys in NE Siberia, (2) a brief analysis of the data set, and (3) a comparison of the field data to remote sensing products. The data are a truly important contribution to forest inventory data in NE Siberia, which are notably scarce considering the region's large extent and importance to global biogeochemical cycles. The data are reported clearly and consistently, and the methods and analyses employed by the authors are appropriate to answer their study questions.

Below, I detail my comments on (1) the dataset and (2) the manuscript.

**Dataset available on PANGEA**

In general, I found the data presentation easy to follow. The metadata are easily accessible, columns are defined, and units are easily findable. However, I suggest the following improvements:

 • The data are currently organized with the metadata in the same file as the tabular data. This is problematic for reading the tabular data into R. I had to spend quite a bit of time figuring out how to delete the top metadata and then reading the tabular data into R correctly. I suggest I suggest having one file for metadata and one file for data, rather than combing them into one file.

Our response:

The data format with this type of header belongs to PANGAEAs policy and is used the same way for all data published there.

When ignoring everything surrounded by the signs "/* … */", the table without the header is more easily readable. Alternatively, the data can be downloaded with "pangear" (an R client for the PANGAEA database (https://github.com/ropensci/pangaear)) or pangaeapy (a python client for the PANGAEA database (https://github.com/pangaea-data-publisher/pangaeapy)).

Reviewer 1 wrote:

 • Within one data file, there are multiple columns with the same name (e.g., 'Plot', 'Tree height', 'Area/locality'), which is needlessly confusing. Some of the column names are very long and include spaces and parentheses (e.g., 'Height quantile [m] (Quantile (25th))'), which can be difficult for working with computer programs. I suggest that data columns be short and unique (e.g., 'Federation', 'District', etc.), or at the very least include an underscore (e.g., 'Location_federation', 'Location_District', etc.).

Our response:

We discussed this with the PANGAEA data reviewer, who had modified our original column names in the process of revising the data. He assured us that the naming of the columns as it is now, was chosen to comply with PANGAEAs policy to bring the data to a standardized format, which aims at making it more findable and interoperable.

Reviewer 1 wrote:

• Some of the comments in the database are in German; could these be translated into English?

Our response:

Thank you for spotting this. English translations were added.

Reviewer 1 wrote:

• The term 'Krumholz' is used in the dataset but not discussed in the text. For clarity, I suggest discussing how growth types (tree vs. krumholz) were determined.

Our response:

We added to the Methodology chapter (2.2):

*...the variable "growth type" can take the values tree (T), shrub (S), tree lying (TL), for lying deadwood, multistem (M), if several shoots emerged from the same base, and krumholz (K). Larix can occur both in the tree form and in the krumholz form. The criterion for the latter is the lack of a straight, upright stem (Kruse et al. 2020b)*

*Kruse, S., Kolmogorov, A. I., Pestryakova, L. A., Herzschuh, U.: Long-lived larch clones may conserve adaptations that could restrict tree line migration in northern Siberia, Ecol Evol. 2020; 10; 10017-10030, https://doi.org/10.1002/ecee3.6660*

Reviewer 1 wrote:

• Were the latitude/longitude coordinates taken plot center or somewhere else? Was the WGS84 datum used to collect the coordinates? This information is important for geolocating in relationship to satellite products.

Our response:

Thank you for this important remark. Yes, the coordinates describe the plot center, and the datum is WGS84. We added to the Methodology (2.2.):

*Geographic coordinates of the plot center were recorded with a GPS device, using the datum WGS84.*

Reviewer 1 wrote:

**Manuscript**
In general, I found the manuscript easy to read and easy to follow. The figures are easily understandable. That being said, the analytical methods (PCA, multilinear regression) need more explanation and in general, the Discussion could benefit from more rigorous analysis. I detail these points in the line-by-line comments below.

Line-by-line comments:

L4: Can you mention how many plots/site?

Our response:

In this study, we generally use the terms "plot" and "site" synonymously, as there was always one plot per site. In section 2.2 we changed one sentence to:

*The exact positioning of the survey plot was finalized on the site, ...*

We hope this formulation makes it a bit clearer.

Reviewer 1 wrote:

L 22: '... with much of these forests located in..''

Our response:

Suggestion adopted

Reviewer 1 wrote:

L36-37: Do you mean there only a few inventories? What kind of 'studies' are you referring to here? There are many more papers that deal explicitly with larch dominated ecosystems of Russia, although maybe they are not inventories as defined here. The addition of a few descriptive words would suffice here.

Our response:

Changed to:

*There are several studies that deal with […], but only few come with forest inventory data.*

Reviewer 1 wrote:

L44-49: While it is true that these aspects are difficult to capture with remote sensing, there have been major advances in this area in the past decade and 'can hardly be captured remotely at all' does not accurately reflect recent advances.

Our response:

We changed it to a weaker wording ('*...are still difficult to capture from space...*'), to do justice to the state of the art, while still making the point.

Reviewer 1 wrote:

L60: remove 'extremely'

Our response:

We replaced it by '*strongly*', because we find it worth pointing out that it is indeed one of the most extreme climates in the world in terms of continentality.

Reviewer 1 wrote:

L61: What kind of exceptions (more or less precipitation?)?

Our response:

The exception from "below 300 mm" is necessarily "more". The reference was also added to Figure 6.

Reviewer 1 wrote:

L107: Which variables? (Please list in this sentence). Also, there should be a flag in the dataset for whether the variable was measured directly or derived, which I do not see.

Our response:

That is true, there is no flag on each variable to tell if it was measured or derived. To compensate for this, we added a table to the appendix. We did not list all derived variables in the text, because there are very many of them, but we added the sentence:

*The entire list, displaying which variables were recorded directly on site, and which were derived from other measurements, can be found in Annex B.*

Reviewer 1 wrote:

L141: "...observations of Landsat images (30 m resolution)."

Our response:

Suggestion adopted

Reviewer 1 wrote:

L144 Specify that the tree cover gain/loss is binary (gained/lost) and not continuous (% lost/gained), as the wording currently implies.

Our response:

See below

Reviewer 1 wrote:

L144-145 'The tree cover loss product is thus derived from the annual forest cover products..." this does not make sense to me, since there is not an annual tree cover product published by Hansen. The only annual product is loss/gain.

Our response:

We changed the entire paragraph to incorporate the two above comments:

*We used the tree cover loss product from the Global Forest Watch project (Hansen et al. 2013) which is based on yearly observations of Landsat images (30m resolution). The project publishes various related data sets, e.g. a product about forest cover gain, and most products are update regularly. The tree cover loss product detects for each pixel if it has been converted from containing tree cover (yes/no) to not containing tree cover, in the time from 2000 to 2019. It assigns the year of the loss to a given pixel, or 0 if no loss has taken place since the year 2000.*

Reviewer 1 wrote:

L154: Delete 'however'

Our response:

You probably meant L149. Suggestion adopted.

Reviewer 1 wrote:

L151: Is the value 100/no data relevant to this analysis? These specifics seem unnecessary.

Our response:

It is indeed not necessary to know that the value is 100 or NA, but it is necessary to know that the data differentiates between disturbed larch forest, undisturbed larch forest and no larch forest.

Reviewer 1 wrote:

L155: More information is needed about how the PCA was conducted. What package in R? Were any data transformations performed prior to ordination?

Our response:

We deleted the PCA from the paper, as it did not contribute much to the understanding of the data.

Reviewer 1 wrote:

L158: More information is needed about how the multilinear regressions were performed. Were variables checked for the assumption of normality? Were variables removed for collinearity? How was the best model selected? Were the residuals checked for homogeneity of variance, etc. Why were univariate regressions performed in addition to multivariate? (I see no reason to perform a univariate regression when multiple predictor variables exist.) Which R packages was used?

Our response:

It is true that we performed none of the mentioned steps which would have been necessary for the rigorous implementation of a multilinear regression model. The multilinear model was seen as a by-product, whose main purpose was to evaluate the information gain that could be obtained by using multiple predictor variables, while the univariate regressions were seen as the main product. (In fact, the information gain is small, as the different predictors are strongly correlated.) The reason why we chose the univariate regressions as main product (presented in Figure 7) was that the objective is more to explore patterns in the data than to actually make robust predictions.

Reviewer 1 wrote:

L162: It is unclear to me what you are assessing with the t-test. As written, it implies you determined the sites with recent tree cover loss with a t-test, but on what? And doesn't the Hansen data show tell you that directly?

Our response:

No, the objective of this t-test was to tell if the two groups (with forest loss and without forest loss according to Hansen) differed significantly in terms of the quotient of living basal area over total basal area. In other words, to see if there is more standing deadwood (as ratio of total basal area) on sites with forest loss. Hopefully, with the additional words in the corresponding results section, it is clearer what we did.

Reviewer 1 wrote:

L169: Are these double inventoried trees flagged in the data? How can we distinguish them?

Our response:

These are trees where the column "Tree, survey protocol" has the value "CIRCLEPLOT".

We added to the Methodology (2.2):

*The variable "survey protocol" tells if the tree was recorded on a rectangular plot ("PLOT"), outside of a plot ("EXTRA"), or on a circular plot. In the latter case, the variable takes the value "PLOTHEIGHT" if only height was measured, and "CIRCLEPLOT" if it is the detailed inventory. Those trees appear twice in the data set, once as "PLOTHEIGHT" and once as "CIRCLEPLOT".*

Reviewer 1 wrote:

L172: delete 'only' and 'while 60 do not.'

Our response:

Suggestions adopted.

Reviewer 1 wrote:

L173: Presenting these numbers as percentages rather than absolute numbers would aid the reader in understanding.

Our response:

We added the percentage in parentheses after the numbers.

Reviewer 1 wrote:

L176: change to 'the two Larix species never occur…'

Our response:

Suggestion adopted.

Reviewer 1 wrote:

L179: delete 'only' and 'up'

Our response:

Suggestion adopted.

Reviewer 1 wrote:

L198: It is not clear to me why these plots were singled out. Are they representative of other plots? Instead of discussing these individual plots, I suggest discussing the meaning of the average Gini coefficient and whether you saw any regional differences in the Gini coefficient. Figure 5 is not discussed at all in the text, and this would be a good place for it.

Our response:

Thank you for this suggestion. We pointed out that the sites for figure 4 were chosen as examples to represent how broadly the height distributions can differ, and added some discussion about the Gini coefficient and its variation with the latitude.

In the image caption:

*The height classes among all species, for six different plots of the Yakutia 2021 expedition, which were chosen as examples for differing height distributions.*

In the Results:

*The Gini coefficients are negatively correlated with the geographic latitude of the plot (Figure 5), but significance and explanatory value of the linear correlation are not high (p-Value 0.021, R2 = 0.33).*

Reviewer 1 wrote:

L212: Re-word to say that Pinus pumila and Salix spp. occur more frequently between 65N and 70N.

Our response:

Re-worded to:

*The taxa Pinus pumila and Salix spec. occur frequently between 65° N and 70° N.*

Reviewer 1 wrote:

L217: Which variables were highly correlated with the first (and second) principal component(s)? As written, the PCA plays a very small role in this paper, so you could consider deleting.

Our response:

PCA was deleted.

Reviewer 1 wrote:

L223: I suggest focusing on the variables with statistically significant relationships rather than the R2, which is likely to be quite low given how noisy ecological field data can be. One idea would be to update figure 7/this section to only include significant relationships, as determined by the properly selected multilinear model. This will draw the reader to which variables are important.

Our response:

All of the relationships except one (mean tree height ~ T01) are significant. We added p-values for each relationship in the graph.

Reviewer 1 wrote:

L227: A multilinear model is more relevant than individual regressions. When properly implemented, you can control for collinearity and report which variables are statistically significant. An R2 of 0.356 is not unreasonable.

Our response:

As discussed above, the multilinear model was seen only as a by-product, and not implemented with preparatory analysis. Nevertheless, we added a correlation matrix for the climate variables into the annex, and referred to it in the section.

An adjusted $R^2$ of 0.356 in itself may be reasonable for a linear regression. But the fact that a model with four predictors has an adj.$R^2$ of 0.356, while using only one of these variables produces adj.$R^2$ = 0.34, shows that there is not much information to be gained by using the other variables. This is the main point which we want to make here.

We also added the sentence:

*In addition to that, multilinear regression models using these four variables are not an advisable prediction tool here, since the predictor variables are correlated among each other (see Appendix D).*

Reviewer 1 wrote:

L240: Appendix is missing an A

Our response:

Corrected accordingly.

Reviewer 1 wrote:

L242: ORNL DAAC is the hosting service. Chen et al. is the correct citation.

Our response:

Thank you. Corrected accordingly.

Reviewer 1 wrote:

L242: Does the field data at these 6 sites indicate there has been a stand replacing fire over this period? (i.e., does the field data corroborate the remote sensing data)?

Our response:

Sometimes, but not everywhere; see below.

Reviewer 1 wrote:

L243: Does the Hansen data set also show losses where there have been stand replacing fires?

Our response:

See changes below.

Reviewer 1 wrote:

L243: This wording is unclear. Do you mean that there are 5 plots where Hansen shows forest loss and Chen at al do not? What do the field data say about those plots?

Our response:

The paragraph was rephrased to:

*The Hansen et al. (2013) data set covers a wider area and different time range. However, there are 5 plots where they detect forest loss in times and places where Chen et al. find that the stand age is at maximum. We encountered clear signs of recent disturbance in the vegetation at only 50\% of the sites where either of the data products detected forest loss.*

Reviewer 1 wrote:

L247: Regarding the plots where dead trees do not contribute a relevant amount of the basal area, was there evidence of logging at these sites? Or perhaps the dead trees are not standing but fallen? This information is helpful in determining how well the Hansen data represent what is happening on the ground.

Our response:

Neither fallen deadwood nor signs of removed trees could explain this. Therefore we added the sentence:

*On most of these, field observations did not find signs of recent disturbance, except for one plot where natural succession was at a pioneer stage.*

Reviewer 1 wrote:

L257: delete 'very'

Our response:

There is no word 'very' in L257. We deleted it in L252.

Reviewer 1 wrote:

270: Here, a distinction should be made between the landscape-scale distribution (all plots combined), which seems to be close-to exponential, and the stand-level distribution (looking at

plots individually), which are not. It makes sense that at the landscape-scale, we would see continuous recruitment, but at the stand-scale, recruitment may be more episodic, likely in response to fires.

Our response:

Thank you for pointing this out, which helped us to refine the sentence as follows:

*This suggests that recruitment patterns are only continuous at the landscape scale, but discontinuous at the local scale, which is consistent with the well-known fact that stand-replacing fires regularly rejuvenate forests in the permafrost ecosystems of our research area.*

Reviewer 1 wrote:

L275: Is there reason to believe that different species should NOT have different allometries? I thought the expectation is that allometries are species-specific.

Our response:

It is true that we expect the allometries for different species to be somewhat different. But the fact that they differ so strongly (i.e. exponents 2.29 for Populus and 0.66 for Abies, for DBS) seems noteworthy.

Reviewer 1 wrote:

L275: You may be able to determine if allometries for larch or pine vary by climate. For example, Berner et al., 2015 found that in some (but not all) species, boreal shrub allometries vary by ecoregion.

Our response:

This is a good idea for further research with the data. A main challenge will be to control the interaction with the stand density, which we already implemented for larch. Pine and spruce, on the other hand, only occur in a small portion of the region we investigated, so that climatic differences between those plots are small.

Reviewer 1 wrote:

L276: What are the sample sizes?

Our response:

We added:

*...the small sample sizes for some species groups, like Abies (10 measurements for DBS) and Populus (27). The species groups with more than 100 measurements (Betula, Larix, Picea, Pinus) have smaller differences among each other in the allometry coefficients.*

Reviewer 1 wrote:

L277: It should be fairly straightforward to calculate the relationship between the derived DBHs and your measured heights in a way that is comparable with data from the literature.

Our response:

In our perception, it would be circular reasoning to use the DBH values that were predicted from the Height to make a Height prediction from DBH. Therefore, the direction in which our prediction goes, impedes a further comparison with most allometries from literature.

Reviewer 1 wrote:

L293: Is this Santoro et al. (2018) a or b?

Our response:

It is a. Corrected accordingly.

Reviewer 1 wrote:

L293: Please give some context for the Santoro et al. (2018) sentence. What data set are they working with or assessing?

Our response:

We added:

*When assessing the reliability of their biomass data product, Santoro et al. (2018a, 2021) note ...*

Reviewer 1 wrote:

L300: What about logging by local communities?

Our response:

At the beginning of the next paragraph, we mentioned that we did not encounter deforestation due to human activities. This does not rule out the possibility of selective logging near settlements, but according to what we saw on expeditions, this only has a small impact.

Reviewer 1 wrote:

L310: Extend = extent

Our response:

Corrected accordingly.

Reviewer 1 wrote:

L315: Reporting results from the multilinear model is more relevant here. Using model selection including variance inflation factors will help you get rid of variables that are collinear.

Our response:

See response below.

Reviewer 1 wrote:

L319: Did you preform proper model selection? It seems that January temp may be too highly correlated with July temp and/or not important enough to be included in a multiple regression model.

Our response:

We addressed some of these issued in the Results section, i.e. by referring to a correlation matrix of the climate variables which we added to the appendix. Here in the discussion section, we deleted the sentence that refers to it, as we did not intend to present the multilinear model as the main product, but rather to explore the relationships between individual variables:

*Thus, we can not conclude that colder winters are favourable for forest growth, *

Reviewer 1 wrote:

L326: Please include a reference for this; Dobricic et al., 2020 or Collow et al., 2022 discuss extreme events, but please also include a reference for increasing drought stress.

Our response:

We added Collow et al. 2022 as a reference for extreme events, and added another sentence with two references for drought stress:

*...growth has been diminished by drought stress and extreme events, which are increasing under climate warming, like the 2020 Siberian heat wave (Collow et al. 2022). Kropp et al. 2017 and Walker et al. 2021 support that water availability is a limiting factor for Larix cajanderi.*

*Collow, A. B. M., Thomas, N. P., Bosilovich, M. G., Lim, Y.-K., Schubert, S. D., and Koster, R. D.: Seasonal Variability in the Mechanisms Behind the 2020 Siberian Heatwaves, Journal of Climate 35.10 (2022): 3075-3090.* https://doi.org/10.1175/JCLI-D-21-0432.1

*Kropp, H., Loranty, M., Alexander, H. D., Berner, L. T., Natali, S. M., Spawn, S. A.: Environmental constraints on transpiration and stomatal conductance in a Siberian Arctic boreal forest. Journal of Geophysical Research: Biogeosciences, 122(3), 487–497 (2017)* https://doi.org/10.1002/2016JG003709

*Walker, X., Alexander, H. D., Berner, L., Boyd, M. A., Loranty, M. M., Natali, S., Mack, M. C.: Positive response of tree productivity to warming is reversed by increased tree density at the Arctic tundra-taiga ecotone. Can. J. For. Res. 51: 1323–1338 (2021).* https://doi.org/10.1139/cjfr-2020-0466

Reviewer 1 wrote:

L332: How can the reader access this additional data?

Our response:

This data is not published yet. We added a remark into the text. (*...other, still unpublished data...*)

Reviewer 1 wrote:

L339: The wording implies that the Potapov data set is expected to be released soon, but the citation says it was published in 2020.

Our response:

The data they published and described in their paper from 2020 does not include the boreal regions north of 52°N, but those are to be published in the future.

Reviewer 1 wrote:

L339: Please describe what the Potapov paper is about.

Our response:

We added the sentence:

*They published a global canopy height data set with 30 m resolution for the tropical and temperate zones of the world, and the data for the boreal regions is to be released soon.*

Reviewer 1 wrote:

L343: aastern = eastern?

Our response:

Yes. Corrected accordingly.

Reviewer 1 wrote:

Figure 2:
-Please include the definition of DBS and DBH in the caption.

Our response:

Adjusted accordingly.

Reviewer 1 wrote:

-It is difficult to see the colors of the different species groups. Can you make the circles in the legend bigger? Perhaps also just a little bigger in the graph.

Our response:

We increased the circle size in the legend. The circle sizes in the graphs were left as they were, to prevent overlapping circles from covering each other up even more.

Reviewer 1 wrote:

-The term 'krumholz' appears in this figure but nowhere in the text. How is it defined here?

Our response:

We added some explanatory words in the Methods section:

*Larix can occur both in the tree form and in the krumholz form. The criterion for the latter is the lack of a straight, upright stem.*

Reviewer 1 wrote:

-Are the allometries here just for larch krumholz and not for larch trees?

Our response:

The two large bottom panels show the allometries for the non-krumholz larches.

Reviewer 1 wrote:

-For the density allometries, are the different colored lines different quartiles? It seems like the bright yellow one (highest density) is not fit to any data (there are no bright yellow points, especially as height increases). It also seems like the high density line (yellow-grey) is below the low density (dark purple) and medium density (purple grey) lines but the highest density (bright yellow) line is above all of them. Why would this be (i.e., why isn't there a negative progression from low to medium to high to highest)?

Our response:

Thank you for bringing this to our attention. The regression lines did not show the actual allometry that was presented in the results, but regressions for subgroups. We changed the lines to represent examples of the actual allometry. We also reduced the number of lines to three, to allow for a better overview, and added more explanation in the caption:

*The regression lines illustrate the allometry for three different stand densities (300, 3000 and 30 000 trees per hectare) while in the actual allometric formula, stand density is a continuous variable.*

It is generally possible that the progression from one line to the other is not monotonous, because the functions varied in factor as well as in exponent. While the factor decreases with stand density, the exponent increases. However, the newly chosen lines look more homogeneous.

Reviewer 1 wrote:

-In caption, can you clarify that the regression lines in the bottom graphs are for illustrative purposes only, because you used stand density as a continuous variable (rather than binned) to calculate DBH and DBS (or at least, this is what I understand from the methods)?

Our response:

See comment above

Reviewer 1 wrote:

Figure 5: This figure is not discussed in the text. Also, please make the legend and entire plot larger for enhanced readability.

Our response:

We performed a linear regression between the variables in the graph, Gini coefficient and Latitude, and added to the results section:

*The Gini coefficients are negatively correlated with the geographic latitude of the plot (Figure 5), but significance and explanatory value of the linear correlation are not high (p-Value 0.021, R² = 0.33).*

Also, the figure size was increased.

Reviewer 1 wrote:

Figure 6: It is not clear to me what this plot adds, and it is not discussed much in the text.

Our response:

You are right that this is not an output which adds much to the results section. However, we find this figure to be helpful for the reader to get an overview over the key climatic data of our research area.

Therefore, we moved it to the front, after Figure 1, since it is first referred to in "2.1. Methodology – Area of Interest"

Reviewer 1 wrote:

Figure 8:
-Please report R2 and p-values for both plots.

Our response:

The original plots did not include the regressions. We added regression lines as well as $R^2$ and p-values for the first plot, but not for the second, as the regression was performed on the non-transformed data.

Reviewer 1 wrote:

- Please discuss (in the text) what it means that the data fit better under the logarithmic transformation.

Our response:

Actually, the data does not fit better under logarithmic transformation: Without log-transformation, GSV captures 25% of the variance in stem volume, and with log-transformation and removal of zeroes, only 23%. Just the bias is lower under the log transformation.

Reviewer 1 wrote:

References

Berner, L. T., Alexander, H. D., Loranty, M. M., Ganzlin, P., Mack, M. C., Davydov, S. P., and Goetz, S. J.: Biomass allometry for alder, dwarf birch, and willow in boreal forest and tundra ecosystems of far northeastern Siberia and north-central Alaska, 337, 110–118, https://doi.org/10.1016/j.foreco.2014.10.027, 2015.
Collow, A. B. M., Thomas, N. P., Bosilovich, M. G., Lim, Y.-K., Schubert, S. D., and Koster, R. D.: Seasonal Variability in the Mechanisms Behind the 2020 Siberian Heatwaves, 1–44, https://doi.org/10.1175/JCLI-D-21-0432.1, 2022.
Dobricic, S., Russo, S., Pozzoli, L., Wilson, J., and Vignati, E.: Increasing occurrence of heat waves in the terrestrial Arctic, 15, 024022, https://doi.org/10.1088/1748-9326/ab6398, 2020

---

## Author Comment (AC2)

We would like to express our gratitude to Anonymous Reviewer 2 for the considerate revision of the paper and the numerous helpful comments.

Below are detailed replies to all the issues raised in the comment.

A revised version of the manuscript has been prepared based on the reviews.

**Reviewer 2 wrote:**

The paper by Miesner et al presents a database of forest surveys distributed across a large area characterized by ecoclimatic gradients in northeastern Siberia. This geographic region has important influences on global carbon and energy dynamics due to its large area and sensitivity to climate change. Despite this importance there is a relative paucity of freely available and easily accessible data that can be used to inform observational and modeling studies. From this perspective, this manuscript describes an important dataset that is worthy of publication and dissemination. Some revision is required before the paper and data can be considered further for publication.

The overall structure of the manuscript is appropriate. The data are described reasonably well, and the comparison with various gridded data products is useful for understanding the utility and limitations of the data set. There are a number of areas where additional detail and/or discussion are warranted. Some of these are noted in my specific comments below, but in general the discussion seems a bit superficial. In particular, it would be useful to have a deeper discussion of errors associated with the use of height as the primary unifying measurement, as well as the visual estimates of height. To my mind, DBH is a more common and useful metric than height, and seems an easier measure than the several required to triangulate height using a clinometer. Related, it seems that there is a high potential for error, that is hard to quantify, associated with the visual estimates of height. More critical discussion here would be nice.

Our response:

We agree that DBH is a more common leading variable than height, and it is often a more robust predictor. As mentioned in the Methodology section, our experience with sporadic control measurements showed that the error of the method is small, although we never quantified it in a systematic way. But when the error propagates, it may still be non-negligible. A reason for choosing height as main variable was that the survey protocol was developed in the tundra-taiga ecotone, where trees are often small in height and have low branches, so that height is not only easier to measure, but also more meaningful. In the more densely forested taiga, this protocol was kept, so that the data would be consistent. We elaborated on this in the Discussion.

Reviewer 2 wrote:

Regarding the data, I found the files a bit unwieldy to work with. There is a lot of awkwardly structured metadata at the top of each file, before the actual data, making it difficult to read the files into a program like R. There also seems to be some redundant data here, in terms of site names, campaign, PI, etc. It may be more appropriate to have separate metadata files in order to make the data more user friendly/analysis ready.

Our response:

The data formatting is part of the PANGAEA data publication and cannot be changed by the authors. There are several ways for dealing with this structure: When ignoring everything surrounded by the signs "/* … */", the table becomes more readable.

Alternatively, the data can be downloaded with "pangear" (an R client for the PANGAEA database (https://github.com/ropensci/pangaear)) or pangaeapy (a python client for the PANGAEA database (https://github.com/pangaea-data-publisher/pangaeapy)).

Reviewer 2 wrote:

These are primarily suggestions - having the data described and available is important, and this paper accomplishes that. The edits/revisions I suggest would improve the utility of the data set.

Below are a number of specific/minor comments, more editorial in nature.

L120: Perhaps Gridded Data Products would be a more appropriate term here. The CHELSA data is downscaled reanalysis/climate data, not a remote sensing data set.

Our response:

We changed it to just "*Data products*" in the heading, and specified in the following sentence:

*...we used several gridded, mostly remote sensing derived data products ...*

Reviewer 2 wrote:

L161: What variable is suitable for comparison with biomass? A little more detail/information here would be nice.

Our response:

Explanation added :

*… like stem volume*

Reviewer 2 wrote:

L242-3: Are the field data consistent with this? Are vegetation conditions consistent with recent disturbance?

Our response:

For both sources, in 50% of the cases where disturbance was detected, the observed vegetation showed clear signs of recent disturbance, while for the rest of the allegedly disturbed sites, it was not clear. We added:

*We encountered clear signs of recent disturbance in the vegetation at only 50% of the sites where either of the data products detected forest loss.*

Reviewer 2 wrote:

L246: Areas with tree loss in the Hansen data set hold more standing dead than those without? Please clarify.

Our response:

See response below

Reviewer 2 wrote:

L247: Plots indicated as having forest loss do not have any disturbed trees? Please clarify.

Our response:

Regarding both above comments, we re-formulated the entire sentence to:

*The average quotient of basal area of living trees to overall basal area is higher for the sites without disturbance than for the sites with forest loss according to the Hansen et al. (2013) data set, which shows that there is more standing deadwood on sites with forest loss.*

Hopefully this makes it more understandable.

Reviewer 2 wrote:

L256: It would be good to discuss in a bit more depth the error implications of visual height estimates. Also, since DBH is a common measure that is often allometrically related to height, biomass, LAI and other ecologically important processes it would be good to discuss the tradeoffs associated with using height instead.

Our response:

We added some sentences stating that we are aware that DBH is the more common measure, but we chose height because it is easier, and to keep the protocol consistent. We are aware that it generates an error which then propagates into other variables, but we did not quantify this error. The paragraph now reads:

*The field work was carried out according to scientific standards. Tree height was chosen as the leading variable because it is easy to overview in sparse stands and it generally correlates well with other variables (stem diameter, biomass). Diameter at breast height (DBH), even though it is more commonly used as a predictor, is more laborious to determine for trees in sparse stands with low crowns. With frequent clinometer measurements, we assured precise height estimations, and the remaining errors can be expected to average out over the high number of observations, which were easily obtained due to the efficiency of the method. Drawbacks coming with this method are: Since the diameter is only predicted from height, errors from this prediction propagate into derived variables like basal area and stem volume. And the initial measurement error, even if small, propagates along the same way. This error was not quantified systematically.*

Reviewer 2 wrote:

L261-4: These sentences are almost too vague to be helpful. What does it mean that the plots are not weighed accordingly? I'm not sure what the last sentence is supposed to mean.

Our response:

*"Weighed accordingly"* meant *"according to the occurrence of the vegetation type they represent"*. We changed it in the text. What we meant to clarify with these sentences is: If we find e.g. a negative correlation between January temperature and basal area, this may be systematic throughout Eastern Siberia, or it may be due to our choice of plots.

Reviewer 2 wrote:

L270: Are there any patterns here, geographic or otherwise?

Our response:

Thank you for raising this very good question.

A quick glance into the data suggests that the sites towards the south are a bit more irregular in their height distributions then the more northerly sites. And indeed, the Gini coefficient for height is negatively correlated with latitude (compare Fig.5), albeit only slightly significantly and with low explanatory value ($R^2 = 0,033$).

We cannot tell, though, if this is due to the generally larger height range on the southern sites, or if it is indeed caused by the disturbance regime, which is known to be characterized by shorter fire return intervals in the south. Still, we added this sentence to the results section:

*The Gini coefficients are negatively correlated with the geographic latitude of the plot (Figure 5), but significance and explanatory value of the linear correlation are not high (p-Value 0.021, R² = 0.33).*

Reviewer 2 wrote:

L271: Are sites recently affected by fire indicated in the database?

Our response:

Unfortunately, no, as this was not thought of at the time of the data publication.

Reviewer 2 wrote:

L288-9: What about variables produced at 30m resolution? How to explain the mismatch for these?

Our response:

Even if a 30m survey plot may not lie exactly inside the pixel, we aimed to take a representative vegetation type for a larger area, so that a scale mismatch should not be the problem here. However, we mentioned other reasons for the mismatch in the following sentences.

Reviewer 2 wrote:

L311: Extend should be extent

Our response:

Corrected accordingly.

Reviewer 2 wrote:

L325: See also papers by Kropp et al and Walker et al for evidence of drought stress in Siberian larch.

Our response:

Thank you for these suggestions. Two of those were added and referred to in an additional sentence, and another one in the sentence before:

*...growth has been diminished by drought stress and extreme events, which are increasing under climate warming, like the 2020 Siberian heat wave (Collow et al. 2022). Kropp et al. 2017 and Walker et al. 2021 support that water availability is a limiting factor for Larix cajanderi.*

*Collow, A. B. M., Thomas, N. P., Bosilovich, M. G., Lim, Y.-K., Schubert, S. D., and Koster, R. D.: Seasonal Variability in the Mechanisms Behind the 2020 Siberian Heatwaves, Journal of Climate 35.10 (2022): 3075-3090.* https://doi.org/10.1175/JCLI-D-21-0432.1

Reviewer 2 wrote:

L349: I didn't think the WorldClim data set was used in this study, please correct/clarify.

Our response:

Thank you for spotting this. It should have been CHELSA; WorldClim was used in an earlier version. Corrected accordingly.

Reviewer 2 wrote:

L355: I'm not sure this conclusion warrants a stand alone paragraph.

Our response:

It is true that the paragraph does not convey any new information, but we liked it as a concise closing statement. It is therefore merged into the preceding paragraph.

Reviewer 2 wrote:

Figure 2 - panels should be labeled a, b, c, etc. and referred to as such in the text.

Our response:

Suggestion adopted.

Reviewer 2 wrote:

Figure 4 - it would be helpful if the figure capture indicated that these specific plots were selected to show examples of different size class distributions.

Our response:

We added to the caption:

*...which were chosen as examples for differing height distributions.*

Reviewer 2 wrote:

Figure 7 - are all of these results significant, and if so to what level?

Our response:

All correlations except the top left (mean tree height ~ T01) are significant, most are striongly significant. We added p-values below the $R^2$-values for each graph.

Reviewer 2 wrote:

Kropp, H., Loranty, M., Alexander, H. D., Berner, L. T., Natali, S. M., & Spawn, S. A. (2017). Environmental constraints on transpiration and stomatal conductance in a Siberian Arctic boreal forest. *Journal of Geophysical Research: Biogeosciences*, *122*(3), 487–497. https://doi.org/10.1002/2016JG003709

Kropp, H., Loranty, M. M., Natali, S. M., Kholodov, A. L., Alexander, H. D., Zimov, N. S., Mack, M. C., & Spawn, S. A. (2019). Tree density influences ecohydrological drivers of plant–water relations in a larch boreal forest in Siberia. *Ecohydrology*, *12*(7), e2132. https://doi.org/10.1002/eco.2132

Walker, X., Alexander, H. D., Berner, L., Boyd, M. A., Loranty, M. M., Natali, S., & Mack, M. C. (2021). Positive response of tree productivity to warming is reversed by increased tree density at the Arctic tundra-taiga ecotone. *Canadian Journal of Forest Research*, cjfr-2020-0466. https://doi.org/10.1139/cjfr-2020-0466

---

## Author Comment (AC3)

We extend our thanks to Anonymous reviewer 3 for making the following remarks:

Reviewer 3 wrote:

Miesner and coauthors compile a data set of forest surveys from expeditions to the north-east of the Russian Federation. Data collection spans a long time period and includes about 10 tree species. They reported forest attributes, such as tree height, DBH, etc. They also compared their data with remote sensing datasets of forest height, biomass, and forest loss, and found the limitations in these remote sensing data. This dataset is invaluable to understand boreal forest conditions and their impacts on high-latitude carbon dynamics. In boreal forests, many tree species are short and are often classified as shrubs. In this work, how did the authors separate forests from shrublands?

Our response:

In our inventories, we did not aim to make a qualified differentiation between forest and other vegetation types like shrublands or forest-tundra. Instead, all sites were considered as forest, as long as any trees grew there, independent of their height or density. The column "Forest type" in the plot data base refers to the predominant growth form of the trees, and not to the vegetation types. Indeed, several sites in our data set do not qualify as forest according to common definitions, due to the low heights of the trees and/or their sparsity, as can be seen in plot data base columns like for example *"Height quantile [m] (Quantile (90$^{th}$))"*, *"Height max [m]"*, *"Tree BA [m**2/ha]"*.

Reviewer 3 wrote:

In the abstract, it is necessary to indicate the time period of data collection.

Our response:

Suggestion adopted:

*We compile a data set of forest surveys from expeditions to the north-east of the Russian Federation, in Krasnoyarsk Krai, the Republic of Sakha (Yakutia) and the Chukotka Autonomous Okrug (59-73° N, 97-169° E), performed between the years 2011 and 2021.*

---

## Referee Report (RR1)

This is the second review of *Forest structure and individual tree inventories of north-eastern Siberia along climactic gradients* by Miesner et al.

The authors addressed almost all of my concerns and, with a few minor revisions, I think the manuscript will be suitable for publication.

General comments:

(1) The use of the terms 'plot' and 'site' interchangeably is unnecessarily confusing because those terms generally mean different things. Please chose one and use it consistently throughout.

(2) The authors note in the main text that multiple regression is not a valid tool because the predictor variables are correlated among each other (line 251). I agree with this assessment, so I wonder why it is included at all in the manuscript. Either multiple regression analysis is appropriate (in which case, more needs to be done to ensure it is being used correctly), or it is not appropriate (in which case, it should not be included in the manuscript). The authors seem to want to have it both ways: they want to use the $R^2$ value from the multiple regression model while simultaneously calling such an analysis 'not advisable.' It is not statistically appropriate to use metrics from an analysis that has not been carried out correctly. I suggest removing all mentions of the multiple regression analysis from the manuscript.

Line-by-line comments:

L64: It is not clear to me that an exception would necessarily be above, as the authors indicated in their response. If this is what you mean, change to 'Annual precipitation is generally below 300 mm, although this is sometimes exceeded towards the boundaries of the area."

L 77: For clarity, change to "the exact position of the survey plots was finalized on-site…."

L156: update should be 'updated'

L223: 'the' is missing…should be 'the significance and explanatory…'

L224: 'value' should be plural

L224: p-value of 0.021 is significant and $R^2$=0.33 is fairly decent for ecological data, so consider removing the part about significance and explanatory values not being high.

L311: In the previous review of this paper, I mentioned that one should be able to rearrange the equation relating DBH to height to compare data from the literature. The authors responded that this would be circular reasoning. Let me be more clear. The authors use the following equation:

$$DBH = a_1(H-1.3)^{a2}$$

We can rearrange this equation to relate height to DBH:

$$H = (DBH/a_1)^{(1/a2)} -1.3$$

However, I briefly looked at the Alexander and Delcourt papers, and I was unable to find evidence that they relate DBH to tree height. Instead, they both relate DBH to biomass. I suggest the authors either compare the results directly or delete this sentence.

---

## Author Response (AR2)

**Author's responses to revievers comments, 2ⁿᵈ round**

essd-2022-152

**Referee report 1 and authors responses**

Referee 1 wrote:
The authors have adequately addressed my previous comments and I have just a few additional suggestions to address before the paper is published. A couple of these have to do with the data organization, and while I see the the pangaear package is convenient for accessing this data, I generally think there could be some improvements still. Specifically, I would recommend including a comment containing a brief description as well as the units for each variable in the parameter list. As it stands, some metadata information is included in the comments for some variables, while in other cases it is included in column headers. This inconsistency can make it hard to find things, and when column headers are used as variable names including acronym definitions and units makes them quite unwieldy. Sorry I didn't catch these things in my previous review.

Our response:
Much of the variable naming in the data tables is due to PANGAEAs policy regarding the organization of data, which aims at using variable names that are more . To help the user to get an overview over the different variables, their meaning and units, we replaced the tables in Appendix B (which until then displayed all variables grouped by whether they are measured or derived). The new table includes the variable names from the *.tab files, the units, acronym definitions, and whether the variables were measured or derived, and if so, from which variable it was derived.

Referee 1 wrote:
The only thing that is more than editorial in nature is the fact that the data set contains repeated observations (mentioned on line 99). It seems that because of this the same tree has multiple unique tree ids - is this correct? I don't think this is tremendously problematic because the plot-level data exists as a separate data set. On the other hand, I could envision a case for example, where a user wants to work directly with the tree level data to calculate other plot-level metrics and aggregates by plot (i.e. the 'Event' variable) thereby including multiple trees. While I realize it us up to the user to read the paper and understand the data, I think that including repeated observations in the data set is not good practice and would recommend either having a more explicit way to indicate which observations are repeated, or publishing the intensive circular plot tree measurements as a separate data set. And on a related note, for the trees included twice, do both records include measured DBH, or is one measured (i.e. CIRCLEPLOT) and the other predicted from height (e.g. PLOTHEIGHT)?

Our response:
Yes, it is true that the same tree can have two entries into the database with two different tree IDs. The two entries can generally not be matched to each other, because the height-only inventory and the detailed inventory were usually performed as two separate steps. Therefore, it is possible that the same tree appears once with measured DBH and once with DBH predicted from height via the allomeries.
In order to mitigate this problem, we have split the database into two tables, which we called "Tree heights", and "Tree Measurements", as the first one contains all entries with the survey protocol "PLOTHEIGHT", and the other one all entries with diameter measurements. The entries with the protocol "PLOT" were copied to be present in both tables. This way, each table for itself does not contain any duplicate trees, "Tree Heights" contains all trees used for the aggregation of variables on the plot level, and "Tree Measurements" contains all entries needed for calculating the allometries.

Referee 1 wrote:
Also related to the data, it is unclear what the difference is between Longitude and Long C as well as Latitude and Lat C are. Appendix B indicates that Lat C and Long C are derived variables, but

the methods or the metadata don't seem to indicate how they were derived or how they differ from plot coordinates acquired in the field.

Our response:
The fields "Lat C" and "Lon C" in the Tree Data Base indicate the coordinates of the plot center, and represent the exact same measurements as the plot coordinates in the Plot Data Base. The reason why they were considered to be "derived variables" in the Tree Data Base is that they were not measured individually for every tree, but copied from the Plot Data Base, and thus derived from the original "measured" variable "Event" in the Tree Data Base. In the new table in the appendix, which replaces the former one that categorized the variables as measured or derived, this is explained as

derived (from Plot Data Base)

Referee 1 wrote:
L151: should be 'updated', I think.

Our response:
Corrected accordingly.

Referee 1 wrote:
L199: I think this sentence refers to basal diameter, but it is a little unclear. Perhaps replace 'It' with 'DBH'.

Our response:
'It' refers to diameter at breast height, as the range of basal diameter was already stated earlier in the paragraph. Adopted accordingly:

*DBH is almost always lower than basal diameter, on average by the factor 0.628. DBH ranges up to …*

Referee 1 wrote:
L235: Is this R2? It would be good to state this explicitly.

Our response:
Adopted accordingly:

*… not exceeding an R² of 0.351 in any combination.*

**Referee report 2 and authors responses**

Referee 2 wrote:
This is the second review of Forest structure and individual tree inventories of north-eastern Siberia along climactic gradients by Miesner et al.

The authors addressed almost all of my concerns and, with a few minor revisions, I think the manuscript will be suitable for publication.

General comments:

(1) The use of the terms 'plot' and 'site' interchangeably is unnecessarily confusing because those terms generally mean different things. Please chose one and use it consistently throughout.

Our response:
We had internally used the term "site" for describing the area of a plot and its immediate surroundings, but since this differentiation was not relevant in the manuscript, they became interchangeable, and we now replaced the term "site" by "plot" throughout the manuscript. The only exception is one occurrence, where there actually is a difference in meaning implied, which is hopefully understandable without further explaination:

*The sites at which the surveys were performed were chosen beforehand with consideration of remote sensing data. […] The exact positioning of the survey plot was finalized on-site, with the aim to have each plot representing a homogeneous vegetation type.*

Referee 2 wrote:
(2) The authors note in the main text that multiple regression is not a valid tool because the predictor variables are correlated among each other (line 251). I agree with this assessment, so I wonder why it is included at all in the manuscript. Either multiple regression analysis is appropriate (in which case, more needs to be done to ensure it is being used correctly), or it is not appropriate (in which case, it should not be included in the manuscript). The authors seem to want to have it both ways: they want to use the R 2 value from the multiple regression model while simultaneously calling such an analysis 'not advisable.' It is not statistically appropriate to use metrics from an analysis that has not been carried out correctly. I suggest removing all mentions of the multiple regression analysis from the manuscript.

Our response:
While the multiple regression may still give some insight about the data, even if its requirements are not fulfilled, we adhered to the suggestion and removed it. This is probably the most appropriate way of dealing with it.

Referee 2 wrote:
Line-by-line comments:

L64: It is not clear to me that an exception would necessarily be above, as the authors indicated in their response. If this is what you mean, change to 'Annual precipitation is generally below 300 mm, although this is sometimes exceeded towards the boundaries of the area."

Our response:
Adopted accordingly.

Referee 2 wrote:
L 77: For clarity, change to "the exact position of the survey plots was finalized on-site…."

Our response:
Adopted accordingly.

Referee 2 wrote:
L156: update should be 'updated'

Our response:
Corrected accordingly.

Referee 2 wrote:
L223: 'the' is missing…should be 'the significance and explanatory…'

Our response:
Sentence deleted (see below).

Referee 2 wrote:
L224: 'value' should be plural

Our response:
Sentence deleted (see below).

Referee 2 wrote:
L224: p-value of 0.021 is significant and R 2 =0.33 is fairly decent for ecological data, so consider removing the part about significance and explanatory values not being high.

Our response:
We re-formulated the paragraph to:

*The Gini coefficents are negatively correlated with the geographic latitude of the plot (Figure 5). The linear regression has a p-Value of 0.021 and R² = 0.33.*

Referee 2 wrote:
L311: In the previous review of this paper, I mentioned that one should be able to rearrange the equation relating DBH to height to compare data from the literature. The authors responded that this would be circular reasoning. Let me be more clear. The authors use the following equation:

DBH = a1*(H-1.3)^a2

We can rearrange this equation to relate height to DBH:

H = (DBH/a1)^(1/a2) -1.3

However, I briefly looked at the Alexander and Delcourt papers, and I was unable to find evidence that they relate DBH to tree height. Instead, they both relate DBH to biomass. I suggest the authors either compare the results directly or delete this sentence.

Our response:
It is true that the mentioned papers do not relate diameters to height, but to biomass. We therefore changed the sentence to:

*There is little literature with which to compare our results, because commonly the diameter is used as predictor variable, and not height, as in Alexander et al. 2012 and Delcourt & Veraverbeke 2022, who both model biomass from diameter.*

But we did not delete the sentence, because we find the two literature references worth citing in this context.
We did not do the rearranging of the equations as suggested, because inverting a DBH~H model generally does not lead to the same results as optimizing a H~DBH model, as the residuals are minimized along different variables.